# The Aerosol Characterization from Polarimeter and Lidar (ACEPOL) airborne field campaign

Kirk Knobelspiesse[1], Henrique M. J. Barbosa[2,11], Christine Bradley[3], Carol Bruegge[3], Brian Cairns[4], Gao Chen[5], Jacek Chowdhary[4,6], Anthony Cook[5], Antonio Di Noia[7], Bastiaan van Diedenhoven[4,6], David J. Diner[3], Richard Ferrare[5], Guangliang Fu[8], Meng Gao[1,9], Michael Garay[3], Johnathan Hair[5], David Harper[5], Gerard van Harten[3], Otto Hasekamp[8], Mark Helmlinger[3], Chris Hostetler[5], Olga Kalashnikova[3], Andrew Kupchock[1,9], Karla Longo De Freitas[1,15], Hal Maring[10], J. Vanderlei Martins[11], Brent McBride[11], Matthew McGill[1], Ken Norlin[12], Anin Puthukkudy[11], Brian Rheingans[3], Jeroen Rietjens[8], Felix C. Seidel[3,10], Arlindo da Silva[1], Martijn Smit[8], Snorre Stamnes[5], Qian Tan[13], Sebastian Val[3], Andrzej Wasilewski[4], Feng Xu[14], Xiaoguang Xu[11], John Yorks[1]

[1]NASA Goddard Space Flight Center, Greenbelt, MD, USA
[2]University of São Paulo, São Paulo, Brazil
[3]Jet Propulsion Laboratory, California Institute of Technology, Pasadena, CA, USA
[4]NASA Goddard Institute for Space Studies, New York, NY, USA
[5]NASA Langley Research Center, Hampton, VA, USA
[6]Columbia University, New York, NY, USA
[7]University of Leicester, Leicestershire, United Kingdom
[8]SRON Netherlands Institute for Space Research, Utrecht, Netherlands
[9]Science Systems and Applications, Inc., Greenbelt, MD, USA
[10]NASA Headquarters, Washington, DC, USA
[11]University of Maryland, Baltimore County, Baltimore, MD, USA
[12]NASA Armstrong Flight Research Center, Edwards, CA, USA
[13]NASA Ames Research Center, Moffett Field, CA, USA
[14]University of Oklahoma, Norman, OK, USA
[15]Universities Space Research Association, Columbia, MD, USA

*Correspondence to*: Kirk Knobelspiesse (kirk.knobelspiesse@nasa.gov)

**Abstract.**

In the fall of 2017, an airborne field campaign was conducted from the NASA Armstrong Flight Research Center in Palmdale, California to advance the remote sensing of aerosols and clouds with Multi-angle Polarimeters (MAP) and Lidars. The Aerosol Characterization from Polarimeter and Lidar (ACEPOL) campaign was jointly sponsored by NASA and the Netherlands Institute for Space Research (SRON). Six instruments were deployed on the ER-2 high altitude aircraft. Four were MAPs: the Airborne Hyper Angular Rainbow Polarimeter (AirHARP), the Airborne Multiangle SpectroPolarimetric Imager (AirMSPI), the Airborne Spectrometer for Planetary EXploration (SPEX Airborne) and the Research Scanning Polarimeter (RSP). The remainder were Lidars, including the Cloud Physics Lidar (CPL) and the High Spectral Resolution Lidar 2 (HSRL-2). The southern California base of ACEPOL enabled observation of a wide variety of scene types, including urban, desert, forest, coastal ocean and agricultural areas, with clear, cloudy, polluted and pristine atmospheric conditions. Flights were performed

in coordination with satellite overpasses and ground based observations, including the Groundbased Multiangle SpectroPolarimetric Imager (GroundMSPI), sun photometers, and a surface reflectance spectrometer.

ACEPOL is a resource for remote sensing communities as they prepare for the next generation of spaceborne MAP and lidar missions. Data are appropriate for algorithm development and testing, instrument intercomparison, and investigations of active and passive instrument data fusion. They are freely available to the public, at 10.5067/SUBORBITAL/ACEPOL2017/DATA001 (ACEPOL Science Team, 2017). This paper describes ACEPOL for potential data users, and also provides an outline of requirements for future field missions with similar objectives.

## 1 Introduction

Aerosols, clouds, and their interactions are the largest source of uncertainty in estimates of the radiative forcing of the Earth. Reducing this uncertainty requires global observations to act as constraints for studies of the role of aerosols and clouds in a changing climate (Boucher et al., 2013). While existing passive orbital sensors show observational skill, they are limited in their consistency and under-determined for retrieval of the relevant geophysical parameters (Mishchenko et al., 2004). In other
words, the remote sensing retrieval solutions are often non-unique, and require the use of constraints in the form of, for example, aerosol models, which may or may not represent geophysical reality. For this reason, the 2007 Decadal Survey of the National Research Council recommended to the National Aeronautics and Space Administration (NASA) the creation of the Aerosol-Cloud-Ecosystems (ACE) Mission, with the stated goal to reduce climate forcing uncertainty of aerosol-cloud interactions, and to better understand ocean ecosystem carbon dioxide uptake (NRC, 2007). This recommendation led to the
ACE pre-formulation mission study (da Silva, et al, 2019). This study (2008-2018) was devoted to technological and scientific developments to address aerosol-cloud climate forcing uncertainty along with improved observations of ocean color to better characterize ocean biology. Designs for an ocean color instrument were enhancements of previous ocean color satellite instruments, such as the Sea-viewing Wide Field-of-view Sensor (SeaWiFS, McClain et al, 2004). These prototypes were, in part, developed and tested by deploying airborne instruments on high altitude aircraft. Thus, airborne field campaigns were a
key component of the ACE pre-formulation study.

ACEPOL was one of several field campaigns supported by ACE, with the specific objective of testing lidar and Multi-angle Polarimeter (MAP) instruments in a variety of conditions. Such instruments reduce aerosol and cloud climate forcing uncertainties by accurately determining aerosol and cloud optical and microphysical properties and vertical distribution. They
are, however, diverse in their measurement characteristics, retrieval approaches, and capability (Weitkamp, 2006, Kokhanovsky et al., 2015, Dubovik et al., 2019). Six of these instruments were installed on the ER-2 aircraft, which, because of its capability for high altitude, long range flights, provides an ideal platform for exploring measurement concepts of relevance to ACE. Two were Lidars: the Cloud Physics Lidar (CPL) and the High Spectral Resolution Lidar 2 (HSRL-2),

while the remaining four instruments were MAPs: the Airborne Hyper Angular Rainbow Polarimeter (AirHARP), the Airborne Multiangle SpectroPolarimetric Imager (AirMSPI), the Airborne Spectrometer for Planetary EXploration (SPEX Airborne) and the Research Scanning Polarimeter (RSP). See Table 1 for a summary of the characteristics of these instruments, and Section 3 for more detail.

In addition to supporting ACE, an ACEPOL objective was to provide a calibration reference for the Cloud-Aerosol Lidar with Orthogonal Polarization (CALIOP) instrument on the Cloud-Aerosol Lidar and Infrared Pathfinder Satellite Observations (CALIPSO) mission (Winker et al., 2009). This was performed with coordinated flights along the satellite ground track at the time of overpass. A coordinated underflight of the Cloud-Aerosol Transport System (CATS) orbital lidar (McGill et al., 2015) were also conducted. The third sponsor of the ACEPOL field campaign was the Netherlands Institute for Space Research (SRON), partly funded through the NWO/NSO project ACEPOL (ALW-GO/16-09), to further advance aerosol measurement capabilities from space and the technological development of the SPEX Airborne instrument in particular.

ACEPOL consisted of nine flights from the Armstrong Flight Research Center (AFRC) in Palmdale, California in October and November of 2017. AFRC is the home base of the ER-2 aircraft and has excellent supporting facilities. Furthermore, it is within range of a variety of types of scenes, from oceans off the California coast, urban areas in Los Angeles, intensive agriculture in the California central valley, forests in the Sierra Nevada mountains and the high desert in California, Nevada and Arizona. Aerosol and clouds within these regions are similarly varied. An additional benefit is the accessibility of many ground validation sites, in particular the aerosol observations of the Aerosol Robotic Network (AERONET, Holben et al., 1998), and desert salt pans such as Rosamond Dry Lake for which comprehensive surface reflectance characterization allows vicarious calibration of airborne sensors. Figure 1 is a pilot's photograph from the cockpit of the ER-2, which illustrates the unique, high altitude vantage point of the aircraft. Deployment of an instrument on this aircraft is a close analog for the space environment and observation conditions. Figure 2 and Figure 3 show the ER-2 on the ground with a portion of the ACEPOL team, and the ACEPOL emblem, respectively. The latter indicates the position of the remote sensing instruments onboard the aircraft, with two on the fuselage, and two in each wing pod.

Consistent with NASA's policy on data collection and availability, ACEPOL data are publicly available (see Section 3.4). The purpose of this article is to document the conditions under which these data were collected to aid their use by the scientific community, and to describe initial efforts by the ACEPOL team to analyze and compare results. As such, Section 2 describes the ACEPOL measurement objectives, while Section 3 covers instrument specifics. Section 4 chronicles the field deployment, and identifies successful observations of targets described in Section 2. Section 5 discusses the value of ACEPOL for various current and planned missions, while Section 6 concludes.

## 2 Objectives

The overall objectives of the ACEPOL field campaign, as is described below, were to test new observations systems, develop new algorithms, and validate orbital observations. For that reason, a wide range of observation conditions were desired. This differs from field campaigns investigating specific processes for the purpose of broader scientific understanding. The goals of this campaign instead focused on improvement of measurement techniques, instrument calibration, and algorithm development. All flights started and ended in Southern California, enabling flights over urban, rural, mountainous, desert, coastal and deep ocean regions in a variety of atmospheric conditions. To better organize flight planning, the ACEPOL measurement objectives were condensed into a list of prioritized targets, as described in Table 2 and below.

Target types fell into four broad categories: calibration, geolocation, validation, and targets of opportunity. Calibration targets (1a, 1b, and 1c) were meant to provide spatially uniform observations with which radiometric and polarimetric measurements between multi-angle polarimeters can be compared. A similar intercomparison was performed during the Polarimeter Definition Experiment (PODEX) between the RSP and AirMSPI instruments (Knobelspiesse et al., 2019). Intercomparison can now be performed with those instruments plus AirHARP and SPEX Airborne. Such intercomparisons can confirm measurement uncertainty estimates and identify calibration problems. Different scene types are useful for intercomparison: cloud free ocean observations (1a) provide low reflectance, potentially highly polarized measurements, while land scenes can have high reflectance but moderate to low polarization. Cloud scenes provide high reflectance and low polarization. Intercomparison requires accurate geolocation, so minimizing scene heterogeneity and atmospheric variation is important. Furthermore, scenes with distinct features, such as coastlines, were targeted to provide a geolocation reference (1d). Validation targets are used to test the geophysical products retrieved by the airborne sensors against similar observations on the ground (2a, 2b), by other field campaigns (2c), and by satellites (3a, 3b). Targets of opportunity are intended for algorithm development and represent infrequently observed or difficult scenes. Finally, it should be noted that the target number designation roughly indicates priority. Low numbered targets are of highest priority and are generally organized such that targets supporting validation or calibration of radiometric quantities have greatest precedence, followed by validation of geophysical products derived from such observations, and then special cases and difficult scenes (targets of opportunity).

Most of the highest priority targets were observed successfully during ACEPOL, although conditions precluded observation of uniform marine stratocumulus cloud decks (target 1c), and most cases of high aerosol loads. The latter was highly unusual for this part of the world, as California's San Joaquin Valley and the Los Angeles metropolitan area are known for typically high aerosol loads. The solution was to overfly controlled forest fire burns farther afield in Arizona. Attempted coordination with the Coupled Air Sea Processes and EM Ducting Research (CASPER) East (Wang et al, 2018) field campaign was unfortunately not possible because of scheduling difficulty and weather. Serendipitously, one flight overlapped with a flight by the Alpha Jet Atmospheric Experiment (AJAX), which carried a payload of atmospheric gas sensors. High aerosol loads

over the ocean (4a) were not observed, but low aerosol load overflights of an AERONET site (2b) on a platform off Long Beach, CA have become the basis for several analysis papers (e.g. Fu et al, 2019, Gao et al, 2020). Another important accomplishment was the successful overflight of Rosamond Dry Lake from multiple headings while a ground based team characterized the spectral reflectance of the lakebed. This was used to vicariously adjust the AirMSPI calibration, and serves as a reference for other measurements as well.

Furthermore, the value of ACEPOL observations exceeds these initial objectives. For example, polarimetric observations of land surfaces may be useful for assessment of Bidirectional Reflectance Distribution Function (BRDF) and Bidirectional Polarization Distribution Function (BPDF) models even by other instruments, and observations over the ocean can be used to help develop ocean remote sensing algorithms (see Section 5.4 for more details on the latter).

## 2.1 The Aerosol Cloud Ecosystems (ACE) mission study

ACE preformulation study activities included defining mission requirements, advancing algorithms and instrument technical readiness by convening science workshops, developing mission design white papers, and supporting new or augmenting otherwise planned field campaigns (see https://acemission.gsfc.nasa.gov). Technological development and scientific utilization of three classes of instruments were supported by the ACE study. These instruments included: cloud and precipitation Radars, atmospheric profiling aerosol and cloud lidars, and MAPs. Lidars and MAPs are the most relevant for aerosol remote sensing and were deployed in early 2013 for PODEX (Diner et al., 2013, Alexandrov et al., 2015, Van Harten et al., 2018, Knobelspiesse et al., 2019) and in ACEPOL during the Fall of 2017. In some ways, PODEX can be considered a precursor to ACEPOL. Both used the high-altitude ER-2 aircraft based at AFRC, flew in a variety of conditions, and deployed AirMSPI, CPL and RSP (among other instruments). PODEX also deployed the Passive Aerosol and Cloud Suite (PACS), an earlier version of AirHARP. ACEPOL offers a larger collection of more mature instruments, notably the SPEX Airborne and AirHARP polarimeters and the HSRL-2 lidar. ACEPOL offers an opportunity to also intercompare single instrument and synergistic algorithms involving lidars and MAPs.

## 2.2 Cloud-Aerosol Lidar and Infrared Pathfinder Satellite Observations (CALIPSO) mission validation

Airborne measurements of particulate backscatter and extinction have been important for assessing CALIOP 532 nm (Powell et al., 2009; Rogers et al., 2011) and 1064 nm (Vaughan et al., 2010; 2019) level 1 attenuated backscatter profiles, level 2 aerosol optical depth (AOD) retrievals (Rogers et al., 2014), aerosol classification methodology (Burton et al., 2013), cirrus cloud properties (Yorks et al., 2011), and combined active (CALIOP) passive (Moderate Resolution Imaging Spectroradiometer, MODIS) retrievals of aerosol extinction profiles (Burton et al, 2010). Using airborne measurements to evaluate CALIOP aerosol backscatter measurements avoids uncertainties caused by systematic errors, spatial inhomogeneities, and distortions associated with using ground based lidar measurements for such validation (Gimmestad et al., 2017).

Consequently, ACEPOL also collected measurements under the CALIOP on flights conducted on October 26[th], November 7th and 9th, and under the CATS sensor on the 19[th] of October.

## 2.3 Netherlands Institute for Space Research (SRON) studies

The ACEPOL campaign served several SRON objectives related to the improvement of MAP instrument performance and aerosol retrievals and investigation of algorithm development for aerosol retrievals using both polarimeter and lidar
measurements. Specifically, the SPEX Airborne instrument deployed during ACEPOL is a prototype of the SPEXone instrument that is contributed to the upcoming NASA PACE mission, due to be launched in late 2022. Validation of the SPEX airborne level-1 data products (radiance and Degree of Linear Polarization, DoLP) during ACEPOL serve to identify possible improvements to be implemented in the SPEXone instrument this is to be contributed to NASA's Plankton-Aerosol-Cloud-ocean Ecosystem (PACE, Werdell et al., 2019) mission. The PACE spacecraft will fly three instruments in low earth orbit.
The primary instrument is a UV-NIR imaging spectrometer with additional SWIR bands, while two MAPs will be contributed: SPEXone, and HARP-2 (see section 3.1.1 for a description of the airborne prototype of that instrument). Additionally, ACEPOL provides a test data set for level-2 algorithm development for SPEXone on PACE. The SRON participation in ACEPOL was funded by the Netherlands Organization for Scientific Research (NWO) and the Netherlands Space Office (NSO).

# 3 Observations

## 3.1 Polarimeters

### 3.1.1 The Airborne Hyper Angular Rainbow Polarimeter (AirHARP)

AirHARP is a wide field-of-view imaging polarimeter designed for characterization of cloud and aerosol optical properties. AirHARP is an amplitude-splitting polarimeter: light entering the front lens is decomposed into three orthogonal linear
polarization states (0, 45, and 90°) by a modified three-way Phillips prism. Each polarization state is imaged by a unique detector array, so the first three parameters of the Stokes vector (I,Q, and, U) are retrieved at pixel by combining co-located information from all three detectors. Stripe filters on the detectors define 120 distinct along track viewing angles across the four HARP wavelengths (spectral widths): 440 (14), 550 (12), 670 (18), and 870 (38) nm, across a total swath of ±57° (±47°) along-track (cross-track). 60 of these angles are at 670nm, specifically designed for studying the structure of the polarized
cloudbow. The high angular resolution (roughly 2˚) enables AirHARP to measure the cloudbow structure at all individual pixels. The remaining three wavelengths have 20 angles each for characterization of aerosols. Figure 4a shows an illustration of the sampling scheme applied by the HARP polarimeters, where each along track viewing angle of the instrument produces a full pushbroom image. The figure also shows an example of data collected during the ACEPOL campaign, its second

deployment. The instrument's maiden field campaign was the Lake Michigan Ozone Study (LMOS) in 2017 (McBride et al., 2020). For ACEPOL, AirHARP collected data over defined targets, as the processing speed of the onboard data acquisition system precluded continuous data collection.

AirHARP is an aircraft demonstration for the HARP CubeSat instrument, a standalone satellite that was successfully launched on 19 February 2020 from the International Space Station (ISS) conducting Earth observations for a year-long mission. A third member of the HARP family is the HARP2 sensor, which will provide global coverage in two days, as part of the NASA PACE mission due to launch no earlier than 2022 (Werdell et al., 2019).

AirHARP Level 0 ACEPOL data are corrected, reconstructed, geolocated, and calibrated in the Hyper-Angular Image Processing Pipeline (HIPP) Level-1B algorithm. These data products are gridded to a horizontal resolution of 2000 pixels per latitude degree and packaged into HDF5 files for distribution in ACEPOL data archive. A Level 2 aerosol retrieval algorithm has been implemented using the Generalized Retrieval for Aerosol and Surface Properties (GRASP) scheme (Dubovik et al. 2011, Puthukkudy et. al 2020). Figure 4 (d) shows the AOD map retrieved using AirHARP measurements and GRASP inversion algorithm for a smoke scene in Figure 4 (c). The magnitude and spatial variability of retrieved AirHARP AOD are in good agreement with collocated lidar (i.e., HSRL-2) observations (Puthukkudy et. al 2020) and are also consistent with the AOD retrieved from AirSPEX and RSP instruments (Fu et al., 2020). Cloud droplet size distribution retrievals are implemented using a traditional parametric fit to Mie scattering curves for liquid water droplets (McBride et al. 2020). Retrievals of ice cloud characteristics, aerosols above clouds, ocean and surface reflectance properties, and atmospheric correction are also important potential uses of HARP data.

### 3.1.2 The Airborne Multiangle SpectroPolarimetric Imager (AirMSPI)

The Airborne Multiangle SpectroPolarimetric Imager (AirMSPI, Diner et al., 2013b) is pushbroom imaging camera used for the characterization of atmospheric aerosols and clouds. In addition to radiometric channels, AirMSPI employs photoelastic modulators (PEMs) to enable accurate measurements of the degree and angle of linear polarization (Diner et al., 2010; van Harten et al. 2018). These polarimetric data help discriminate between different aerosol particle types, which is crucial to improving our understanding on climate and air quality. The instrument flies aboard NASA's high-altitude ER-2 aircraft, and acquires Earth imagery with ~10 m spatial resolution across a 9km wide swath. Radiance data are obtained at 355, 380, 445, 470, 555, 660, 865, 935 nm. The polarimetric channels at 470, 660, and 865 nm report both radiances and the linear polarization Stokes components Q and U. AirMSPI is a precursor to the future Multi-Angle Imager for Aerosols (MAIA) satellite instrument (see Section 5.3), which will be used to improve our understanding of the health risks associated with airborne particulate matter.

The AirMSPI camera is mounted on a motorized single-axis gimbal to enable multiple views of a science target from different along-track view angles. The number of views and specific set of angles for each observing sequence is programmable prior to flight. While approaching the target in a straight and level flight line, the pilot presses one of three buttons to start the corresponding multi-angle acquisition sequence. Figure 5 shows example imagery of the sequences used during the ACEPOL campaign. All 3 modes begin and end with views of the onboard dark target and polarization validator light source.

AirMSPI observed a total of 24 targets in the 9-angle step-and-pseudostare mode, 32 targets in the 15-angle mode, and 209 continuous sweep images. AirMSPI L1B2 geolocated radiometric and polarimetric data and quicklooks are available as noted in Section 3.4. For a further analysis, and comparison with other MAPs and Lidars during ACEPOL, see Fu et al, 2020.

### 3.1.3 The Research Scanning Polarimeter (RSP)

A pair of RSP instruments (denoted RSP1 and RSP2; the latter was used in ACEPOL) have been deployed on more than 25 field missions in the last 20 years. The RSP is an airborne multi-angle and multi-spectral polarimeter that continuously scans in the aircraft along-track direction. During a complete scanner rotation, a total of 205 samples, each with a 0.8° (14 mrad) field of view, are collected. The 205 samples include: 152 views between 60° forward and aft of the normal to its baseplate; 10 samples viewing an internal dark reference and 43 samples through an earth viewing polarization scrambler. Samples are obtained by a rotating polarization compensated mirror assembly with six bore-sighted telescopes, to simultaneously obtain Stokes parameters 9 spectral bands. Each telescope uses a Wollaston prism to split the incoming intensity into two spatially separated orthogonally polarized components, which are then further split and passed through dichroic filters, defined by the spectral bandpass: (full width half maximum bandwidths in parentheses) 410.3 (30), 469.1 (20), 555.0 (20), 670.0 (20), 863.5 (20), 960.0 (20), 1593.5 (60), 1880.0 (90) and 2263.5 (120)nm. On the ER-2, the un-vignetted viewing angle range is from 65° aft to 45° forward, for a total of 120 samples. The single nadir ground pixel size from an altitude of 20 km is 280 m, and successive pixels partially overlap.

RSP2 was calibrated in the Airborne Sensor Facility at NASA Ames Research Center before and after ACEPOL. Radiometric calibration was stable to within roughly 1% for all bands, except for 410.3 nm, where a 4% decrease in radiometric throughput was observed (polarimetric calibration was stable to within ~ 0.1%). Data processing to level 1b (calibrated and geolocated at-sensor measurements) consists of dark subtraction for each scan and application of calibration coefficients to generate Stokes parameters, together with the geolocation of each viewing angle sample. Processing from level 1b to level 1c consists of collocating the different viewing angles about a defined altitude, either at the ground, or at cloud top if a cloud is present. The latter requires cloud identification and a cloud top height estimate (Sinclair et al. 2017). Level 2 processing of cloudy scenes is split between water and ice clouds. Water cloud retrievals include cloud optical depth and standard bi-spectral droplet size estimates (Platnick et al. 2017), as well as both parametric (Alexandrov et al. 2012a) and non-parametric (Alexandrov et al. 2012b) droplet size distribution estimates that use polarization (Alexandrov et al. 2018). Most clouds observed during

ACEPOL were low level water clouds, but when ice clouds were detected, cloud optical depth, particle size and particle shape/roughness retrievals (Van Diedenhoven et al. 2012, 2013; Van Diedenhoven 2018) were retrieved. Level 2 processing for aerosol retrievals uses the Microphysical Aerosol Properties from Polarimetry (MAPP) algorithm (Stamnes et al. 2018). The MAPP land surface model consists of a Fresnel reflectance with shadowing for the polarized reflectance (Waquet et al. 2009) together with a RossThick (Ross, 1981, Roujean et al. 1992) LiSparse (Li and Strahler, 1992 ) kernel model (Wanner et al. 1995), similar to that used for operational processing of MODIS land surface products (Schaaf et al., 2002). RSP was operational for all ACEPOL flights and data are available at the ACEPOL data archive (Section 3.4). Example RSP observations of a liquid phase cloud are shown in Figure 6.

### 3.1.4 The Airborne Spectrometer for Planetary EXploration (SPEX Airborne)

The SPEX airborne instrument (Smit et al., 2019) employs the spectral modulation technique (Snik et al, 2009), in which the degree and angle of linear polarization are encoded in a modulation of the radiance spectrum as a function of wavelength. This modulation is achieved by placing a set of dedicated optical components (quarter wave retarder, multiple order retarder) in front of the telescope. The resulting two light beams that contain a modulation pattern as a function of wavelength (out of phase with each other) that enter a spectrometer and are focussed on a detector. Radiance measurements are obtained between 400-800 nm at the spectral resolution of the spectrometer (~2 nm), while for DoLP the spectral resolution is determined by the modulation period, which is conservatively estimated at 10 nm at 400 nm and 40 nm at 800 nm.

An example of a SPEX airborne radiance and DoLP measurement obtained during ACEPOL is shown in Figure 7.

SPEX airborne has nine viewports that are projected on one detector, at angles (+/-57˚, +/-42˚, +/-28˚, +/-14˚ and 0˚). The SPEX airborne cross-track swath is 6º. The data processing from level-0 (detector counts) to level-1b (calibrated and geo-located radiance and DoLP values) consists of the following steps: Dark image subtraction, spectral extraction and line-of-sight annotation, wavelength annotation, spectral alignment of the 2 modulated spectra, radiometric correction, demodulation, and geolocation. Next, data of the different viewports are all aggregated on the same spatial grid, yielding the level 1C data product, currently computed at 1x1 km$^2$ spatial resolution.

For level-2 processing (Fu et al., 2020), the SPEX airborne team has focused on aerosol retrievals, building further on the SRON aerosol retrieval algorithm previously used for POLDER-3 processing (Hasekamp et al., 2011, Lacagnina et al., 2016;2017) and groundSPEX (van Harten et al, 2014; di Noia et al, 2014). The algorithm has been extended from a bi-modal retrieval scheme to a multi-mode retrieval scheme for an arbitrary number of modes (Fu and Hasekamp, 2018). For processing ACEPOL data, a setup with 5 modes has been used, where for each mode the aerosol column number is retrieved corresponding to a set of fine- and coarse mode spectrally varying refractive indices, the fraction of spherical particles (Dubovik et al, 2006), and the aerosol layer height. SPEX airborne data have been processed using 16 discrete wavelengths between 450 and 750

nm. Wavelengths less than 450 nm and greater than 750 nm are excluded because of lower data quality. SPEX airborne was operational for all ACEPOL flights, and data are available at the ACEPOL data archive.

## 3.2 Lidars

### 3.2.1 The Cloud Physics Lidar (CPL)

The Cloud Physics Lidar (CPL) is a nadir-pointing, multi-wavelength (355, 532, 1064 nm) elastic backscatter lidar that measures vertical profiles of cloud and aerosol properties (McGill et al. 2002). CPL has participated in numerous field campaigns since its first deployment in 2000 on the ER-2, often in tandem with the other remote sensing instruments involved in ACEPOL. Raw CPL data is calibrated by normalizing the 355, 532 and 1064 nm signal to the Rayleigh (molecular) backscatter between 15 and 18 km, creating vertical profiles of the total attenuated backscatter (McGill et al., 2007). CPL data products retrieved from the calibrated backscatter include vertical profiles of depolarization ratio, particulate backscatter coefficient, and particulate extinction coefficients, as well as layer properties such as top/base altitudes, aerosol type, and optical depth (Yorks et al., 2011; Hlavka et al., 2012).

CPL data products, with vertical resolution of 30 m and 1 sec frequencies (~200 m horizontal resolution), enable a wide-range of applications including the analysis of cloud properties (McGill et al., 2004; Bucholtz et al., 2010; Yorks et al., 2011; Alexandrov et al., 2015), as well as aerosol and dust properties (McGill et al., 2003; Nowottnick et al., 2011; Wu et al., 2016). CPL data from ACEPOL are fully processed and data products available at the ACEPOL data repository. However, an electronics controller failed in the CPL laser during the second flight of the campaign. The cause of this failure was not immediately apparent, and there was no time to take the instrument out of service during the ACEPOL campaign. Thus, the ACEPOL CPL data products after the first flight are non-standard, as they required more averaging, sometimes as much as 10 sec (2 km horizontal), and removal of some data at specific wavelengths due to poor quality. Nevertheless, data are useful for validation of CALIPSO (McGill et al, 2007, Yorks et al., 2011b, Hlavka et al., 2012) and CATS data (Yorks et al., 2016, Pauly et al, 2019.)

### 3.2.2 The High Spectral Resolution Lidar (HSRL-2)

HSRL-2 is the second generation airborne High Spectral Resolution Lidar developed at NASA Langley Research Center. Like the first generation HSRL-1 (Hair et al., 2008), HSRL-2 independently measures aerosol backscatter and extinction (532 nm) using the HSRL technique, aerosol backscatter (1064 nm) using the standard backscatter technique (Müller et al. 2014, Sawamura et al., 2017), and aerosol depolarization at both wavelengths. HSRL-2 adds measurements of aerosol backscatter and extinction at 355 nm using the HSRL technique as well as measurements of aerosol depolarization at 355 nm. These HSRL-2 measurements are used to compute AOD as well as the aerosol extinction/backscatter ratio ('lidar ratio') at 355 and

532 nm. Rogers et al. (2009) evaluated the HSRL extinction coefficient profiles and found that the HSRL extinction profiles are within the systematic errors of airborne in situ measurements at visible wavelengths. Derived products include, first, estimates of planetary boundary layer heights which use the vertically resolved profile measurements of aerosol backscatter to derive aerosol mixed layer heights during the daytime (Scarino et al., 2014). Second, aerosol classification, which uses an algorithm to interpret the information about aerosol physical properties indicated by the HSRL-2 aerosol intensive parameters to qualitatively infer aerosol type (Burton et al., 2012). HSRL-2 measurements have also been used to retrieve height-resolved parameters such as aerosol effective radius and concentrations (number, surface, volume) (Müller et al. (2014), Sawamura et al., 2017).

HSRL-2 has been deployed on various NASA aircraft for both NASA and DOE field missions (Berg et al., 2016; Sawamura et al., 2017; Burton et al., 2018). During ACEPOL, HSRL-2 operated autonomously from the NASA ER-2 aircraft. When operated from the ER-2, the nominal HSRL-2 aerosol backscatter profiles are reported at a vertical resolution of 15 m and a horizontal/temporal resolution of 1-2 km (10 seconds). Aerosol depolarization ratios at all three wavelengths are reported at the same resolutions. For ACEPOL, aerosol extinction, AOD, and lidar ratio from the HSRL methodology are not available in some cases, particularly when the aerosol loading was small. In these cases, the aerosol extinction at 355 nm and 532 nm is derived using the aerosol backscatter and an assumed lidar ratio of 40 *sr* and reported at the backscatter resolution. In other cases where the HSRL method is available for extinction products, they are reported at 150 m vertical resolution and at temporal resolution of 60s generally and 10s within smoke plumes. Calibrated aerosol backscatter derived using the HSRL method is available in all cases. Problems with the ER-2 coolenol pump and a circuit breaker caused HSRL-2 data gaps on the October 27 flight and loss of HSRL-2 data on the November 1 flight.

## 3.3 Ground observations

Ground based observations are valuable in that they can be used to validate or calibrate aircraft instrument measurements, or provide context for those observations. While not part of the ACEPOL field campaign, ground sites of two instrument networks served as targets for overflights. These networks (the Aerosol Robotic Network, or AERONET, and air quality monitors of the California Air Resources Board, or CARB) host their data in separate archives that are not under direct control of ACEPOL participants. A third effort, also ground based, involved deployment of radiometric and atmospheric measurement at Rosamond Dry Lake. The collection of these data were funded by ACEPOL and are archived at the locations in Table 3.

## 3.3.1 Aerosol Robotic Network (AERONET) sites

The Aerosol Robotic Network (AERONET, Holben et al, 1998) is a system of sun photometers used to produce global measurements of AOD, intensive properties (e.g. refractive index, size distribution), and precipitable water. This NASA product provides data recorded every 15 minutes utilizing many spectral bands, and are available at the AERONET archive (aeronet.gsfc.nasa.gov). During ACEPOL, the Modesto, Fresno-2, CalTech, and Bakersfield sites in California were

operational, and were routinely targeted. Additional AERONET sites that were targeted during the campaign are the USGS Flagstaff site in Arizona and the USC SeaPrism site. The latter is part of a sub-network designated AERONET-Ocean Color (AERONET-OC). This network comprises enhanced instruments that have the ability to determine water-leaving radiance in addition to aerosol optical properties (Zibordi et al, 2009).

### 3.3.2 California Air Resources Board (CARB) sites

Air particulates are measured by the California Air Resources Board (CARB) throughout the state. A majority of CARB sites report PM2.5 (particulate matter with an aerodynamic diameter of less than 2.5 microns) and PM10 (particulate matter with a diameter of less than 10 microns), as determined by a variety of measurements, including particle counting and mass spectrometry. Gaseous criteria for air pollutants that are regulated in California and at the national level are also measured and reported to ensure compliance with U.S. Environmental Protection Agency (U.S. EPA) air quality standards, and are stored at

the CARB archive (https://www.arb.ca.gov/adam/index.html).

In addition to CARB total PM monitors, several speciated EPA PM monitors are located in California's San Joaquin Valley (SJV) including Modesto, Fresno, Visalia, and Bakersfield. The speciated PM monitors in Modesto, Fresno and Bakersfield are collocated with AERONET instruments making these sites very useful for determining connections between near-surface

PM loading and atmospheric aerosols. Modesto, Fresno, and Bakersfield sites were regularly targeted throughout the campaign.

### 3.3.3 Rosamond Dry Lake ground instrumentation

Vicarious calibration (VicCal) is the process of calibrating an on-orbit or aircraft sensor by observation of an Earth target which is characterized for its optical properties. A radiative transfer program is then used to propagate the surface

measurements into a radiance incident on the flight sensor. Ground measurements made in support of ACEPOL vicarious calibration were performed on October 25, 2017, at Rosamond Dry Lake (34.858704º N, 118.07638º W). This included measurements of surface reflectance and total column optical depth at the time of overflight, roughly 18:00 UTC. Data were processed at JPL and are hosted in the ACEPOL archive. These data were used by the AirMSPI team to derive a multiplier needed to adjust the laboratory-derived radiometric gain coefficient. For the Rosamond VicCal, the ratio of radiances using

the laboratory calibration to the VicCal measured radiances was found to be low for the UV bands (0.85), but within 2% of unity for the remaining visible bands. The discrepancy in the UV lands is under investigation, but one error candidate is the low light level for these bands during the laboratory calibration.

The VicCal campaign at Rosamond included measurements of surface reflectance and total column optical depth at the time

of overflight, roughly 18:00 UTC. Results of these measurements are shown in Figure 8 and Figure 9. AOD was measured by two Microtops and a Reagan sunphotometer. The Microtops are handheld instruments, with spectral bands at 440, 675, 870,

938, and 1020 nm. The Reagan sunphotometer (Bruegge et al. 1992; Shaw et al. 1973), built at the University of Arizona, is an auto-tracking sunphotometer, with channels at 370, 400, 440, 520, 620, 670, 780, 870, 940, and 1030 nm. The instrument was calibrated via the Langley method by taking early morning data the day before the ER-2 overflight. A discrepancy is noted in the Microtops data, perhaps due to an error in its surface pressure. As the Reagan instrument was calibrated in-situ, its output was considered higher in accuracy, and used in the AirMSPI VicCal. Also shown in Figure 8 is the Junge model fit to the Microtops data, which assumes a linear relationship on a log-log scale of aerosol optical depth versus wavelength. Surface reflectance was measured by the Analytical Spectral Devices (ASD) FieldSpec 4 instrument. This measures from 350 to 2500 nm with 1 nm samples with resolution at 3 nm (VIS, NIR) and 10 nm (SWIR). Data were taken over a 500x500 m area, along with spectra of a Spectralon 100% diffuse reflectance standard. To report radiances, the Spectralon data are interpolated at the target sample times, and corrected for Spectralon bi-directional reflectance. The ratio of the target to interpolated Spectralon data provides a per-sample reflectance, which are then averaged to provide a site average.

Operating alongside the vicarious calibration instrumentation was the Ground-based Multiangle SpectroPolarimetric Imager (GroundMSPI) instrument (see Fig. 8). The specifications of the GroundMSPI camera are similar to AirMSPI (see Section 3.1.2), including the 8 spectral bands within 355-935 nm, with 470, 660, and 865 nm being polarimetric. The camera is mounted ~1m from the ground on a motorized altazimuth tripod. GroundMSPI performed continuous elevation scans at different azimuths to image the surface as well as the clear sky. The surface observations provide a direct measurement of the polarized bidirectional reflectance distribution function (p-BRDF), including potential spectral variance, whereas the sky radiance and polarization data serve as input for aerosol retrievals, either independently or combined with the airborne sensors. On November 7, 2017, GroundMSPI performed sky scans at the Fresno AERONET station during ER2 and Cloud-Aerosol Lidar and Infrared Pathfinder Satellite Observations (CALIPSO) overpasses. GroundMSPI L1B2 rectified and co-registered radiometric and polarimetric data and quicklooks are available as noted in Section 3.4.

## 3.4 Data availability

The primary repository for ACEPOL data is the NASA Atmospheric Science Data Center (ASDC), at the Langley Research Center. AirMSPI and GroundMSPI data are stored separately at the ASDC, while AERONET data are located at their own archive. Air quality from California are stored at the CARB website. AJAX data are available by request to the PI, Dr. Laura Iraci at NASA ARC. The DOI for the primary database is: 10.5067/SUBORBITAL/ACEPOL2017/DATA001 (ACEPOL Science Team, 2017), while for AirMSPI it is 10.5067/AIRCRAFT/AIRMSPI/ACEPOL/RADIANCE/ELLIPSOID_V006 and 10.5067/AIRCRAFT/AIRMSPI/ACEPOL/RADIANCE/TERRAIN_V006 (ACEPOL AirMSPI Science Team, 2017). GroundMSPI data are at 10.5067/GROUND/GROUNDMSPI/ACEPOL/RADIANCE_v009 (GroundMSPI Science Team, 2017. Table 3 lists further details of these archives.

## 4 Deployment

### 4.1 Flight planning

Daily flight planning was informed by weather and aerosol forecasts from NASA's Global Modeling and Assimilation Office (GMAO, https://gmao.gsfc.nasa.gov) and the European Center for Mid-Range Weather Forecast (ECMWF, http://ecmwf.int), satellite imagery from several geostationary and polar orbiting satellites (e.g., GOES-16, MODIS, VIIRS), and fire weather outlooks from NOAA's Storm Prediction Center (https://www.spc.noaa.gov/products/fire_wx). Flight plans were drafted based on forecast information and adjusted in real-time to account for the rapid changes in fire and weather conditions. See

Table 4 for detailed information about each flight. This table lists relevant information about each flight, including observed targets, instrument status, and coordinated observations with satellite, ground or other aircraft observations. Local time at AFRC is UTC-08:00. Scenes of particular interest are highlighted in bold. The "Moving Lines" flight planning tool (LeBlanc, 2018) was used to prepare for all flights, along with weather forecasting support of the NASA Global Modeling and Assimilation Office (GMAO).

### 4.2 Aircraft operations

All airborne sensors were deployed together on the NASA Lockheed Earth Resources (ER-2) aircraft, based at the NASA AFRC in Palmdale, California. The ER-2 is a high-altitude aircraft, capable of flying up to roughly 21km (68,000 feet), above most of the Earth's atmosphere. Flights were contained within the continental United States and offshore, primarily in California, Nevada and Arizona. The aircraft altitude, range, and the variety of conditions within that range made the ER-2

deployment from AFRC an excellent orbital analogue platform.

Figure 10 Flight tracks for the ACEPOL field campaign.is a graphical illustration of ACEPOL flight tracks with more information in Table 4. Generally, atmospheric conditions during ACEPOL were clear, with very low aerosol loads, and few clouds. The range of the ER-2 helped compensate for these somewhat unusual low aerosol conditions, providing for cloud

observations in northern California or offshore, and deployments east over Arizona to overfly prescribed forest fire burns near the Grand Canyon in Arizona.

One of the priorities of ACEPOL was coordinated overflights of ground sites to validate retrievals by the remote sensing instruments. AERONET sun photometers were overflown on all days except the test flights on October 19th and November

3rd. Often, these overflights were planned so that the aircraft heading was close to the Solar Principal Plane (SPP), which maximizes the scattering angle range of MAP observations in order to increase the measured information. The AERONET-OC site "USC_SEAPRISM", located on an oil platform off Newport Beach, California, was of particular interest because of the capability for concurrent ocean and atmosphere measurements. This site was targeted on multiple days. Since CARB

particulate matter monitoring sites are often co-located with AERONET sites (especially in the California Central Valley), these were frequently monitored as well.

On the last ACEPOL flight (November 9[th]), there was a serendipitous overflight of a (lower altitude) flight by an Alpha Jet stationed at the NASA Ames Research Center (ARC) as part of the Alpha Jet Atmospheric Experiment (AJAX) flight series (Hamill et al, 2016). The Alpha Jet was instrumented with a wing pod sampling gas concentrations of $CO_2$, $CH_4$, $H_2O$, $O_3$ and HCHO, plus 3D wind speeds, temperature and pressure, and proceeded the ER-2 by about one hour in an area of the eastern Sierra Nevada Mountains.

### 4.3 Example scene

ACEPOL brings together four multi-angle polarimeters and two lidars which have disparate observation characteristics. An example of this is shown in Figure 11. This scene, an overflight of prescribed burns near the Grand Canyon on October 27, 2017, was observed by all instruments, although CPL and RSP data are omitted from this figure for clarity. Aerosol Backscatter in the lidar data (HSRL shown, CPL omitted) show smoke in the process of either being lofted by surface topography or trapped by that topography. Large scale AirHARP data indicate the horizontal extent of the smoke plume, while high spatial resolution AirMSPI data demonstrate fine scale spatial variability. Continuous sampling polarimeters (SPEX Airborne, shown, RSP, omitted) capture the atmospheric state for the entire flight. All instruments can be used to further derive the aerosol optical properties to varying degrees of success, depending on observation information content and retrieval algorithm design. Because of the overlapping nature of such observations, these scenes provide a test of retrieval capability (e.g. Fu et al., 2020, Gao et al., 2020, McBride et al., 2020, Puthukkudy et al, 2020) . They can also be used to investigate algorithms that incorporate data from multiple instruments.

## 5 Value for future missions

### 5.1 Aerosol, Cloud, Convection and Precipitation (ACCP) mission study

ACEPOL was designed to evaluate instrument designs under consideration for the ACE mission study, which had scientific goals that are still relevant today. As evidence, the 2017 Decadal Survey of the National Academies of Sciences, Engineering, and Medicine (National Academies of Sciences, Engineering, and Medicine, 2018) identified targeted observables ("Aerosols" and "Clouds, Convection & Precipitation"). In response, NASA is now conducting the Aerosol, Cloud, Convection and Precipitation (ACCP) pre-formulation study (https://science.nasa.gov/earth-science/decadal-accp). Spaceborne versions of the airborne lidar and polarimeters that participated in ACEPOL are under consideration for ACCP. Hence, ACEPOL data play an important role in ACCP activities.

## 5.2 Hyper Angular Rainbow Polarimeter (HARP) CubeSat

AirHARP provides an excellent proxy dataset for the development of HARP CubeSat algorithms. The two instruments are nearly identical, although AirHARP experienced temperature, vibrational, and humidified conditions on aircraft that are not present in space. Like AirHARP, the HARP CubeSat does not have an on-board calibrator, so many of the correction techniques used on the AirHARP data will carry over to the CubeSat data. However, by pointing the spacecraft, the HARP CubeSat sensor is capable of lunar, limb, and deep space views, as well as coverage over a larger variety of surfaces, which may improve post-processing and vicarious data calibrations.

The synergy between AirHARP and other ACEPOL instruments provides excellent opportunities to study design and sampling, with relevance to HARP Cubesat. Because of altitude and bandwidth limitations, the instruments have different spatial resolutions. For AirHARP, ground pixel is much smaller (55m) than it will be from HARP CubeSat (about 4km at nadir). AirHARP retrievals can be used to study small-scale variabilities in a cloud field (McBride et al. 2020) and smoke plume (Puthukkudy et. al 2020). These features will be much less resolved in HARP CubeSat data, which is better suited for intercomparisons with MODIS and other satellite instruments. As technology demonstration, the amount of data collected by HARP CubeSat is severely constrained by spacecraft storage and downlink capabilities allowing for a single scene per day. This limitation is not present in future versions of the HARP payload, such as HARP2 on PACE.

## 5.3 Multi-Angle Imager for Aerosols (MAIA)

NASA selected the Multi-Angle Imager for Aerosols (MAIA) investigation (Diner et al., 2018) as part of the third Earth Venture Instrument (EVI-3) solicitation. MAIA's primary objective is to assess the impacts of different types of airborne particulate matter (PM) on human health, where "type" refers to the proportions of aerosols having different sizes, shapes, compositions and speciated particulate matter counds. The satellite instrument contains a pushbroom multispectral and polarimetric camera mounted on a two-axis gimbal. Along-track pointing (up to ±60º) enables multiangle observations over a discrete set of globally distributed target areas, while cross-track pointing provides the camera with a wide cross-track field of regard (±45º), permitting frequent revisits of the designated targets. MAIA makes use of the same dual-PEM polarimetric imaging approach as AirMSPI and extends the spectral range into the shortwave infrared (SWIR). The camera includes 14 spectral bands in the ultraviolet (UV), visible/near-infrared (VNIR), and SWIR, of which 3 are polarimetric (442, 645, and 1040 nm). Channels near the $O_2$ A-band (749, 762.5 nm) are included to explore sensitivity to aerosol layer (and cloud) height. Launch into a 740-km sun-synchronous, ascending node orbit with 10:30 am equator-crossing time is planned for mid-2022 aboard the General Atomics Orbital Test Bed-2 spacecraft. MAIA measurements will be used to retrieve total AOD and column effective aerosol optical and microphysical properties at 1-km spatial resolution, using an optimal estimation algorithm (Xu et al., 2017). A geostatistical regression modeling framework will be used to transform column aerosol optical and microphysical properties to speciated, near-surface PM concentrations (Kalashnikova et al, 2018). During ACEPOL, AirMSPI collected

imagery over speciated PM and AERONET sites, while HSRL-2 obtained independent data regarding aerosol types and their vertical distributions, making the campaign a valuable source of information for testing and validation of MAIA algorithms.

### 5.4 Plankton, Aerosol, Cloud, ocean Ecosystem (PACE) mission

The NASA PACE mission, due to be launched in late 2022, will contain three instruments. The primary instrument is the Ocean Color Imager (OCI), a UV-VIS spectrometer designed for Ocean Color remote sensing applications. OCI also has
channels in the SWIR, and will also perform retrievals of aerosol, cloud, and land surface geophysical properties. The PACE payload will also contain two, contributed, MAPs. Airborne prototypes of these instruments, with similar characteristics, were flown as part of ACEPOL. AirHARP is the airborne prototype of PACE/HARP2, while SPEX Airborne is the prototype of PACE/SPEXone (Werdell et al, 2019, Hasekamp et al., 2019). For these reasons, ACEPOL data provide a valuable resource for the development of remote sensing strategies for PACE, by acting as a proxy for the MAPs on PACE. The spectrometer
characteristics of SPEX Airborne can also serve as a proxy for OCI UV-VIS spectrometer, while the SWIR channels on RSP (with some exceptions) can stand in for those on OCI. Furthermore, the characteristics of other MAPs deployed during ACEPOL can be used to understand the impact of PACE design decisions. Additionally, coincident observation by other instruments, namely the CPL and HSRL-2 lidars, can be used to validate assessed retrievals. Studies of these observations in the context of PACE are active and underway (e.g. Smit et al., 2019, Fu et al, 2020).

### 5.5 Upcoming SPEX Airborne deployments

As part of the EU-Horizon 2020 project SCARBO (Space Carbon Observatory) it is planned that SPEX airborne will participate in an airborne campaign in September 2020 (depending on the COVID-19 situation), flying together with two instruments that measure CO2. The goal is to use aerosol measurements from SPEX airborne to improve CO2 retrievals from 2 instruments dedicated to CO2 retrieval: the MAMAP (Methane Airborne Mapper) spectrometer (Krings et al., 2011) and the
NanoCarb. interferometer (Ferrec et al, 2019) The flights will depart from Toulouse and will cover (parts of) France, Spain, Italy and Germany. The aircraft for the SCARBO campaign will be a Falcon that will fly between 2 and 10 km altitude. For upcoming deployments, the data-processing chain (from level 0 to level-2) will make full use of the algorithm processor performed for ACEPOL (Smit et al., 2019; Fu et al., 2020).

### 6 Conclusions

The ACEPOL field campaign explored techniques for remote sensing of aerosol and cloud optical properties with a variety of MAP and lidar designs. Roughly four categories of observations were targeted: those that aid in radiometric property calibration or validation, those supporting geolocation tests, those for validating the retrieval of derived geophysical parameters, and 'targets of opportunity' representing difficult or rarely observed scenes to use in algorithm development. The field campaign was largely a success: all instruments were operational with limited outages for technical or engineering

problems, and data have been processed and archived. Conditions were sufficient to achieve most of the primary objectives, although unusually low aerosol loads compelled the research aircraft to target forest fires further afield.

These co-located observations, gathered in a diversity of conditions, are a valuable resource for algorithm development, instrument design, and other studies, and are archived in publicly available databases. Furthermore, the coordination and
teamwork demonstrated during this successful field campaign can serve as a model for future campaigns, especially those that have multiple objectives, instrument teams and funding institutions.

## 7 Acronyms and units

| | | |
|---|---|---|
| | ABI | Advanced Baseline Imager |
| | ACCP | Aerosol, Cloud, Convection and Precipitation |
| 550 | ACEPOL | Aerosol Characterization from Polarimeter and Lidar |
| | AERONET | Aerosol Robotic Network |
| | AERONET-OC | Aerosol Robotic Network – Ocean Color |
| | AFRC | NASA Armstrong Flight Research Center |
| | AirHARP | Airborne Hyper Angular Rainbow Polarimeter |
| 555 | AirMSPI | Airborne Multiangle SpectroPolarimetric Imager |
| | AJAX | Alpha Jet Atmospheric Experiment |
| | AOD | Aerosol Optical Depth |
| | ARC | NASA Ames Research Center |
| | ASD-AC | Airborne Science Data for Atmospheric Composition |
| 560 | ASDC | Atmospheric Science Data Center (at NASA Langley Research Center) |
| | BPDF | Bidirectional Polarization Distribution Function |
| | BRDF | Bidirectional Reflectance Distribution Function |
| | CALIOP | Cloud-Aerosol Lidar with Orthogonal Polarization |
| | CALIPSO | Cloud-Aerosol Lidar and Infrared Pathfinder Satellite Observations |
| 565 | CARB | California Air Resources Board |
| | CASPER | Coupled Air Sea Processes and EM Ducting Research |
| | CATS | Cloud-Aerosol Transport System |
| | CPL | Cloud Physics Lidar |
| | DISCOVER-AQ | Deriving Information on Surface Conditions from COlumn and VERtically Resolved Observations |
| 570 | Relevant to Air Quality | |
| | DOE | Department of Energy |

| | | |
|---|---|---|
| | DoLP | Degree of Linear Polarization (unitless) |
| | ECMWF | European Center for Mid-Range Weather Forecast |
| | EPA | U.S. Environmental Protection Agency |
| 575 | ER-2 | Lockheed Earth Resources 2 (aircraft) |
| | EVI | Earth Venture Instrument |
| | GroundMSPI | Groundbased Multiangle SpectroPolarimetric Imager |
| | GMAO | Global Modeling and Assimilation Office |
| | GOES-16 | Geostationary Operational Environmental Satellite – 16 |
| 580 | GRASP | Generalized Retrieval for Aerosol and Surface Properties |
| | HARP2 | Hyper Angular Rainbow Polarimeter – 2 (contribution to PACE mission) |
| | HDRF | Hemispherical-Directional Reflectance Factor |
| | HIPP | Hyper-Angular Image Processing Pipeline |
| | HSRL-2 | High Spectral Resolution Lidar 2 |
| 585 | LaRC | NASA Langley Research Center |
| | LMOS | Lake Michigan Ozone Study |
| | MAIA | Multi-angle Imager for Aerosols |
| | MAMAP | Methane Airborne Mapper |
| | MAP | Multi-angle Polarimeter |
| 590 | MAPP | Microphysical Aerosol Properties from Polarimetry (RSP algorithm) |
| | MISR | Multi-angle Imaging SpectroRadiometer |
| | MODIS | Moderate Resolution Imaging Spectroradiometer |
| | NASA | National Aeronautics and Space Administration |
| | NIR | Near infrared |
| 595 | NIST | National Institute of Standards and Technology |
| | NSO | Netherlands Space Office |
| | NWO | Nederlandse Organisatie voor Wetenschappelijk Onderzoek (Netherlands Organization for Scientific Research) |
| | ORACLES | ObseRvations of Aerosols above CLouds and their intEractionS |
| 600 | P-3 | Lockheed P-3 Orion (aircraft) |
| | PACE | Plankton, Aerosol, Cloud, ocean Ecosystem |
| | PACS | Passive Aerosol and Cloud Suite |
| | PM | Particulate Matter |
| | PODEX | Polarimeter Definition Experiment |
| 605 | RSP | Research Scanning Polarimeter |

| | | |
|---|---|---|
| | SCARBO | Space Carbon Observatory |
| | SeaBASS | SeaWiFS Bio-optical Archive and Storage System |
| | SeaWiFS | Sea-viewing Wide Field-of-view Sensor |
| | SJV | San Joaquin Valley |
| 610 | SPEX Airborne | Airborne Spectrometer for Planetary Exploration |
| | SPEXone | Spectrometer for Planetary Exploration (contribution to PACE mission) |
| | SPP | Solar Principal Plane |
| | sr | Steradian |
| | SRON | Netherlands Institute for Space Research |
| 615 | SWIR | Shortwave Infrared |
| | TCAP | Two Column Aerosol Project |
| | ViCal | Vicarious calibration |
| | VIIRS | Visible Infrared Imaging Radiometer Suite |
| | VIS | Visible (wavelengths) |

## 8 Author contribution

The ACEPOL field campaign is the product of a large team, many of whom are co-authors of this manuscript. KK lead the creation of the manuscript. HM, AdS and FCS conceived and supported the ACEPOL field campaign from NASA HQ. OH coordinated support of ACEPOL at NWO/NSO. RF and KK performed flight planning and coordinated the campaign in the field. AdS and KLDF provided meteorological support for field operations. KN lead ground and aircraft support at AFRC. JVM is the PI for AirHARP, DJD is the PI for AirMSPI, BC is the PI for RSP, OH is the PI for SPEX Airborne, MM is the PI for CPL, and JH is the PI for HSRL-2. CB lead the ground characterization at Rosamond Dry Lake. GC coordinates the data archive. All authors provided material for the manuscript and overall review.

## 9 Competing interests

The authors declare that they have no conflict of interest.

## 10 Acknowledgements

Funding for the ACEPOL field campaign came from NASA (ACE and CALIPSO missions) via the Radiation Sciences Program and the NWO/NSO project ACEPOL (Aerosol Characterization from Polarimeter and Lidar) under project number
ALW-GO/16-09. See Sections 2 for more details.

The ACEPOL field campaign would not have been possible without the professionalism and energy of the ER-2 pilots and ground crew, and the overall support at the NASA Armstrong Flight Research Center.

Research at the Jet Propulsion Laboratory, California Institute of Technology was carried out under a contract with the National Aeronautics and Space Administration (80NM0018D0004).

We thank the United States Forest Service, in particular Jason Clawson and David Hercher, for providing information about planned prescribed burns in the Kaibab National Forest.

The Principal Investigator for the USC_SEAPRISM AERONET-OC site is Curtiss O. Davis of Oregon State University. The site manager is Matthew Ragan of the University of Southern California.

Figure 11 was visualized using Google Earth.

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

**Tables**

**Table 1 Airborne instrument characteristics as implemented on the ER-2 during ACEPOL. *AirMSPI has polarization sensitivity for 470, 660 and 865nm only, all other polarimeters have polarization sensitivity in all spectral channels.**

| Polarimeters | Principal Investigator | Spectral band centers | View angles | Spatial sampling | Spatial resolution |
|---|---|---|---|---|---|

| | | | | | |
|---|---|---|---|---|---|
| AirHARP | J. Vanderlei Martins, UMBC | 4: 440, 550, 670, 870 nm | 670nm: 60 in ±57° fore to aft of nadir, other channels: 20 in ±57° fore to aft of nadir | ±47° cross track FOV, targeted sampling mode, (~42.9km at ground) | Gridded to 2000 pixels per latitude degree, native value: (~55m) |
| AirMSPI | David J. Diner, JPL | 8: 355, 380, 445, 470*, 555, 660*, 865*, 935 nm | Varies with targeting mode, 10 or 15 angle pseudo-stare or continuous sweep | ±15° cross track FOV (9km at ground), targeted sampling mode. | 10m at ground for pseudo-stare, 25m for continuous sweep |
| RSP | Brian Cairns, NASA GISS | 9: 410.3, 555, 469.1, 670, 863.5, 960, 1593.5, 1880, 2263.5 nm | 120 from 45° fore to 65° aft of nadir | Single pixel (280m), continuous sample | 280m, partial successive pixel overlap |
| SPEX Airborne | Otto Hasekamp, SRON | 400-800 nm, ~2nm resolution for intensity, 10-40 nm for polarization | 9: +/-57°, +/-42°, +/-28°, +/-14° and 0° | 6° cross track, continuous sample | Native resolution ~200m (nadir) - 1km (+/-57°). |
| **Lidars** | | **Backscatter channels** | **Extinction Channels** | **Horizontal resolution** | **Vertical resolution** |
| CPL | Matthew McGill, NASA GSFC | 355, 532, 1064 nm | - | ~200m | 30m |
| HSRL-2 | Chris Hostetler, NASA LaRC | 355, 532, 1064 nm | 355, 532nm | 1-2km | 15m |

**Table 2 Prioritized ACEPOL measurement targets. In this table, low aerosol loading indicates a mid-visible Aerosol Optical Depth (AOD) less than 0.1, moderate AOD between 0.1 and 0.2, and high AOD greater than 0.2.**

| Target | Description | Achieved? | Dates (2017) |
|---|---|---|---|
| 1a | Calibration over ocean with no clouds or aerosols | Yes | 10/23, 10/25, 11/07 |
| 1b | Calibration over land with no clouds or aerosols | Yes | 10/25 |
| 1c | Calibration over spatially uniform cloud deck | Partially | 11/01 |
| 1d | Geolocation using coastlines with no clouds | Yes | 10/23, 10/25, 10/26, 11/07 |
| 1e | Coordinated CALIOP/CALIPSO or CATS underflight | Yes | 10/19, 10/26, 11/07, 11/09 |
| 2a | Validation with AERONET with medium to high aerosol loading | Yes | 10/23, 10/25, 10/26, 11/01, 11/07, 11/09 |
| 2b | Validation with AERONET with low aerosol loading | Yes | 10/23, 10/25, 10/26, 11/01, 11/07, 11/09 |
| 2c | Validation against CASPER field campaign | No | None, but one overlap with an AJAX flight on 11/09 |
| 3a | Satellite intercomparison for aerosol retrievals | Yes | 10/23, 10/27, 11/01 |
| 3b | Satellite intercomparison for cloud retrievals | Partially | 11/09 |
| 4a | Targets of opportunity: high aerosol loads over ocean | No | - |
| 4b | Target of opportunity: high aerosol loads over land | Yes | 10/27, 11/01, 11/07 |
| 4c | Targets of opportunity: multiple aerosol layers | No | - |

| | | | | | |
|---|---|---|---|---|---|
| 5 | Targets of opportunity: aerosol above cloud | | No | - | |
| 6 | Targets of opportunity: high aerosol loads over urban surfaces | | No | - | |
| 7 | Targets of opportunity: marine stratocumulus clouds far from land | | No | - | |
| 8 | Targets of opportunity: broken clouds far from land | | No | - | |
| 9 | Targets of opportunity: low clouds over land | | Yes | 11/01, 11/03 | |
| 10 | Targets of opportunity: Cirrus clouds | | Yes | 10/19, 10/23, 11/03, 11/07, 11/09 | |

**Table 3 ACEPOL data availability. Archives for the primary ACEPOL instruments (ASDC, AirMSPI and GroundMSPI) contain calibrated Level 1 data with direct physical observations, e.g. radiance, degree of linear polarization, backscatter. Level 2 data are products of retrieval algorithms and include, for example, aerosol intensive and extensive parameters. These may or may not exist**
**in the database(s) depending on the retrieval algorithm maturity.**

| Archive | URL, DOI |
|---|---|
| ASDC | https://asdc.larc.nasa.gov/project/ACEPOL<br>10.5067/SUBORBITAL/ACEPOL2017/DATA001 |
| AirMSPI | https://eosweb.larc.nasa.gov/project/airmspi/airmspi_table<br>10.5067/AIRCRAFT/AIRMSPI/ACEPOL/RADIANCE/ELLIPSOID_V006<br>10.5067/AIRCRAFT/AIRMSPI/ACEPOL/RADIANCE/TERRAIN_V006 |
| GroundMSPI | https://eosweb.larc.nasa.gov/project/airmspi/airmspi_table<br>10.5067/GROUND/GROUNDMSPI/ACEPOL/RADIANCE_v009 |
| AERONET | https://aeronet.gsfc.nasa.gov/ |
| CARB | https://www.arb.ca.gov/adam/index.html |

**Table 4 Daily description of ACEPOL flights. Nine flights, with a total of 40.1 flight hours, were carried out between October 19, 2017 and November 9, 2017 for ACEPOL.**

| Flight Date | Takeoff (UTC) | Duration (hours) | Targets achieved | Instrument status | Coordinated observations | Notes |
|---|---|---|---|---|---|---|
| 10/19 | 16:09 | 2 | **1e**, CATS: 17:32<br>**10**: 18:00 | All instruments functioning | CATS, AERONET | Test flight recalled early due to high winds at landing site. Because of recall, missed potential coordination with CASPER in situ sampling in Central Valley. |
| 10/23 | 17:01 | 5.7 | **1a**, late in flight<br>**1d**, San Francisco Bay, 19:27<br>Catalina Island, 21:07<br>**2a**, Bakersfield: 17:50, 18:40<br>**2b**, Modesto: 19:20<br>USC_SeaPRISM: 21:07<br>**3a**, Terra: 18:40<br>**10**, 19:54-20:26 | All instruments functioning | Terra (MODIS & MISR) AERONET, AERONET-OC | Severe clear (few clouds, low aerosol load) flight with satellite coordination. **Over flight of ocean AERONET USC_SeaPRISM site has ideal geometry for polarimeters, subject of comparison studies.** |
| 10/25 | 16:30 | 5.9 | **1a**, near coast, 20:49<br>**1b, 5 legs over Rosamond Dry lake: 17:18, 17:50, 18:21, 18:47, 19:24**<br>**1d**, near coast, 20:49, Salton Sea 21:17<br>**2a** CalTech: 18:25, Fresno: 20:15,<br>**2b** USC_SeaPrism:18:30, Bakersfield: 19:45 | All instruments functioning | GroundMSPI at Rosamond Dry Lake (34.85636N, 118.07649W), AERONET, AERONET-OC | Severe clear, primary focus was **overflights of Rosamond Dry Lake** while a ground team characterized surface reflectance. |

| | | | | | | |
|---|---|---|---|---|---|---|
| 10/26 | 18:00 | 4.5 | **1d**, S. California coast: 19:40, Salton Sea: 19:53<br>**1e**, CALIPSO underflight: 20:50<br>**2a, Fresno: 18:48**<br>**2b, Bakersfield: 19:11** | All instruments functioning | CALIPSO, AERONET | **Central valley AERONET overflights,** followed by CALIPSO track coordination. Cloud free |
| 10/27 | 17:00 | 3.2 | **3a** Terra: 18:21<br>**4b** Smoke from fires: 18:00, 18:32, 18:55 | All instruments functioning, some HSRL gaps (ER-2 cooling problem) | AERONET Terra (MODIS & MISR) | Flight targeting prescribed burns in Arizona. Cloud free. |
| 11/01 | 16:35 | 5 | **1c** S. California coast: 17:30<br>**2a** Bakersfield: 17:00<br>**2b** Flagstaff: 19:20<br>**3a** Terra: 18:39<br>**4b** Smoke from fires: 19:12<br>**9** Los Angeles basin: 17:45 | No HSRL (ER-2 cooling problem), only one AirMSPI target, Reduced AirHARP observations | AERONET Terra (MODIS & MISR) | Marine stratocumulus clouds off S. California, then smoke in Arizona. |
| 11/03 | 18:58 | 2.9 | **9** 20:00, 20:42<br>**10** 21:10 | All instruments functioning | | Short flight to test aircraft/instrument repairs. California central valley with multi-layered clouds. |
| 11/07 | 17:19 | 5.3 | **1a** 20:31<br>**1d** Monterey Bay: 20:16<br>**1e** CALIPSO underflight, California Central Valley: 21:18<br>**2a** Bakersfield: 17:55, 19:06, 21:57, Fresno: 19:40, Modesto: 20:06<br>**2b** CalTech: 18:10<br>**10** 18:21, 20:40 | AirHARP lost heater, no data for end of flight | CALIPSO, AERONET | Targeting AERONET sites and CALIPSO track, plus clear segment over the ocean. |
| 11/09 | 17:16 | 5.6 | **1e** CALIPSO underflight, California and Nevada desert, 21:07<br>**2a** Fresno: 22:02<br>**2b** Flagstaff: 18:24, 19:55<br>**2c** AJAX flight overlap, 21:50<br>**4b Smoke from fires: 19:36, 19:52**<br>**10** end of flight | All instruments functioning | CALIPSO, AERONET, AJAX | **Return to Arizona for smoke near the Grand Canyon, successfully observed high aerosol loads.** CALIPSO under flight, coincident measurement with AJAX flight. |

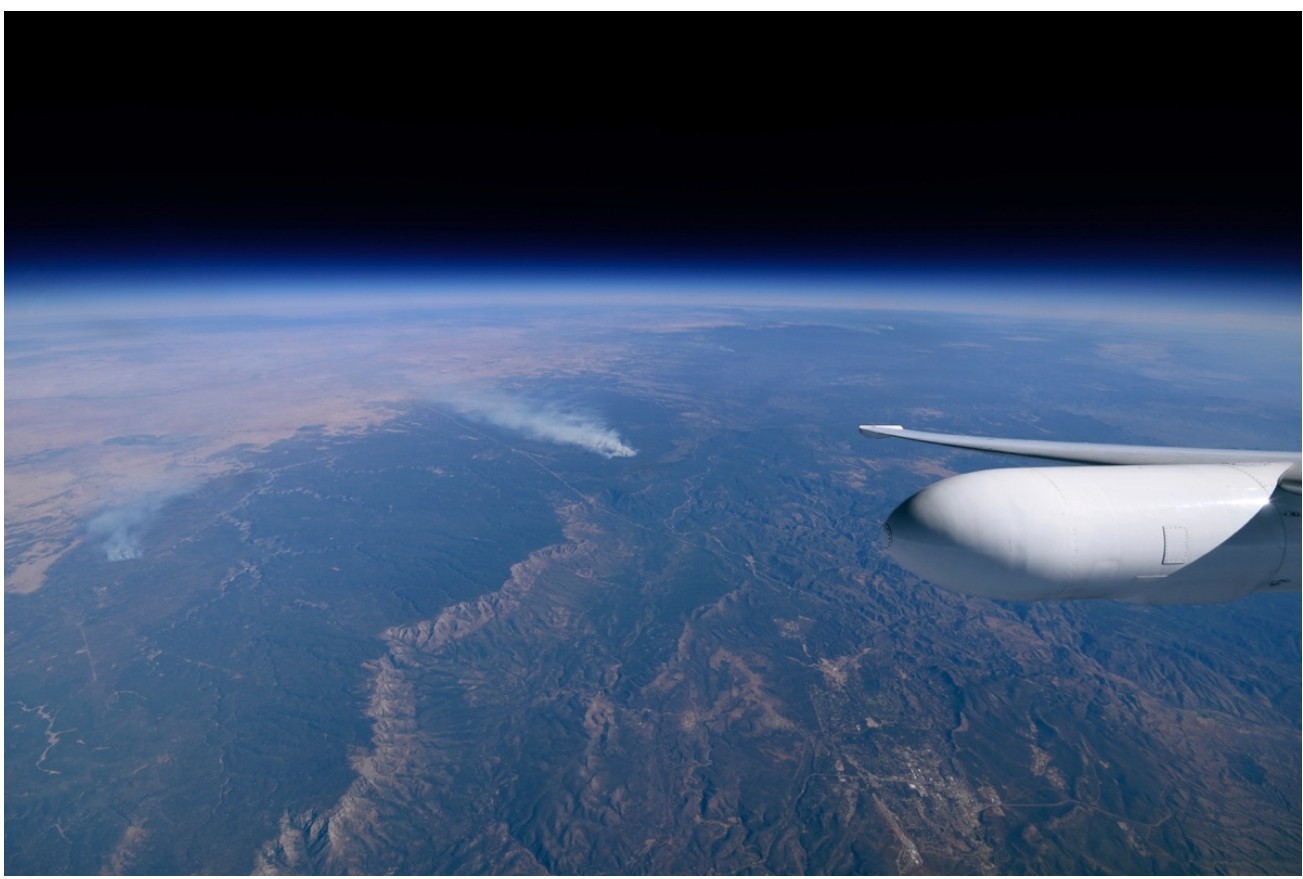

**Figure 1 Photograph from the ER-2 of smoke from prescribed burns in Kaibab National Forest in Arizona on November 9th, 2017. Credit: NASA/Stu Broce.**

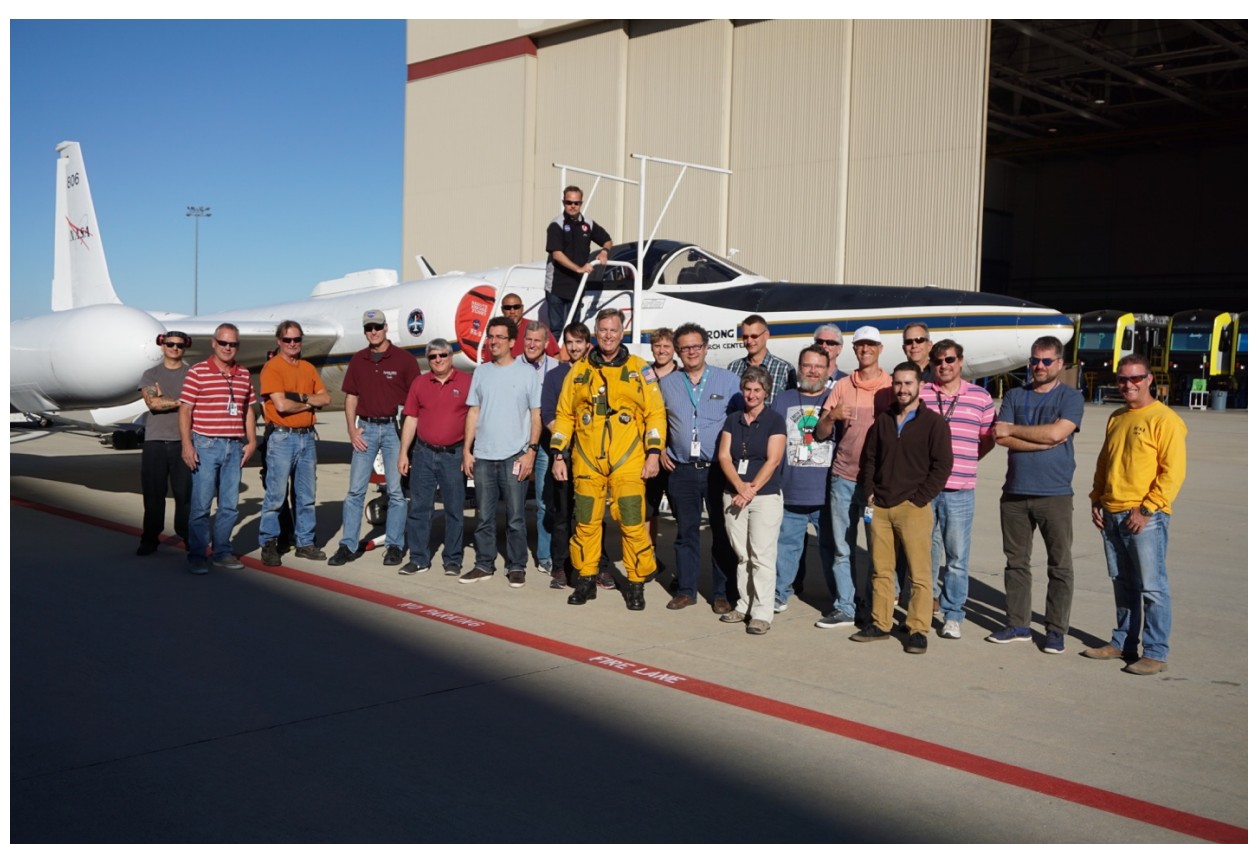

**Figure 2 A portion of the ACEPOL team with the NASA ER-2 at the NASA Armstrong Flight Research Center in Palmdale, California.**

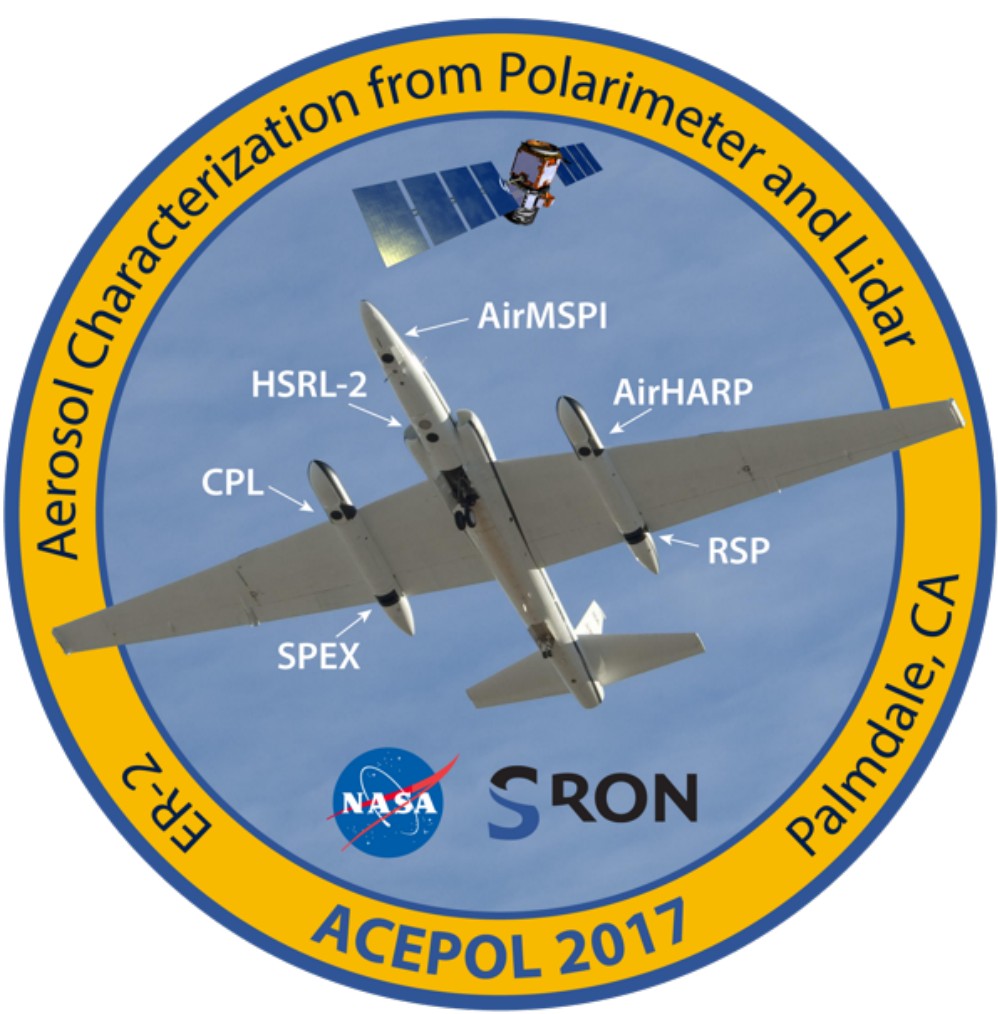

**Figure 3 The ACEPOL field campaign emblem, which also shows the positions of instruments onboard the ER-2 aircraft.**

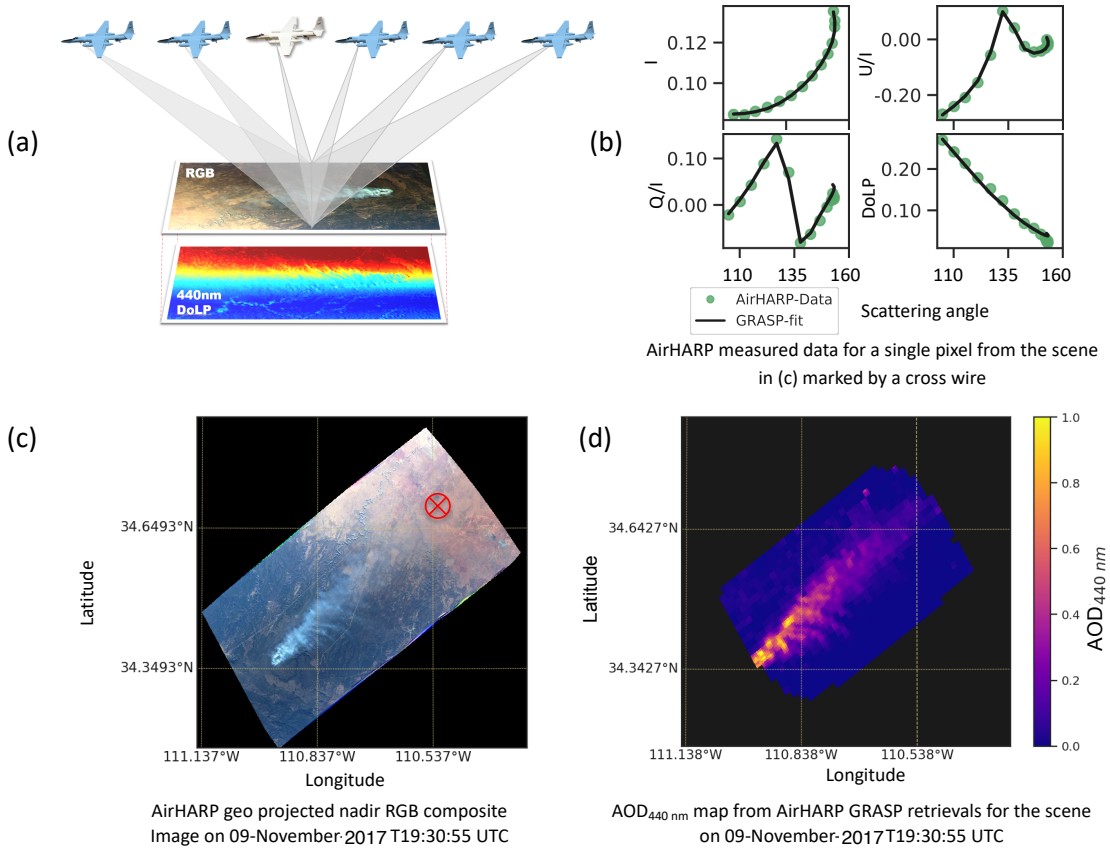

AirHARP measured data for a single pixel from the scene in (c) marked by a cross wire

AirHARP geo projected nadir RGB composite Image on 09-November·2017 T19:30:55 UTC

AOD$_{440\,nm}$ map from AirHARP GRASP retrievals for the scene on 09-November·2017 T19:30:55 UTC


**Figure 4 (a) Illustration of the AirHARP sampling scheme applied during the ACEPOL experiment. Each along track viewing angle produces a full pushbroom image. Six along track viewing angles are shown in the subplot. HARP has up to 60 along track viewing angles for 670nm and up to 20 viewing angles for other wavelengths; (b) AirHARP measured (solid circles) I, Q/I, U/I and DoLP for a pixel marked by red cross wire in (c), plotted as a function of scattering angle and GRASP forward model fit for the variables of same pixel (solid black lines); (c) RGB composite image of a smoke scene collected on the 9th of November, 2017 at 19:31 UTC; (d) Map of aerosol optical depth (AOD) at 440nm retrieved for the same scene using AirHARP measurements and the GRASP inversion algorithm.**


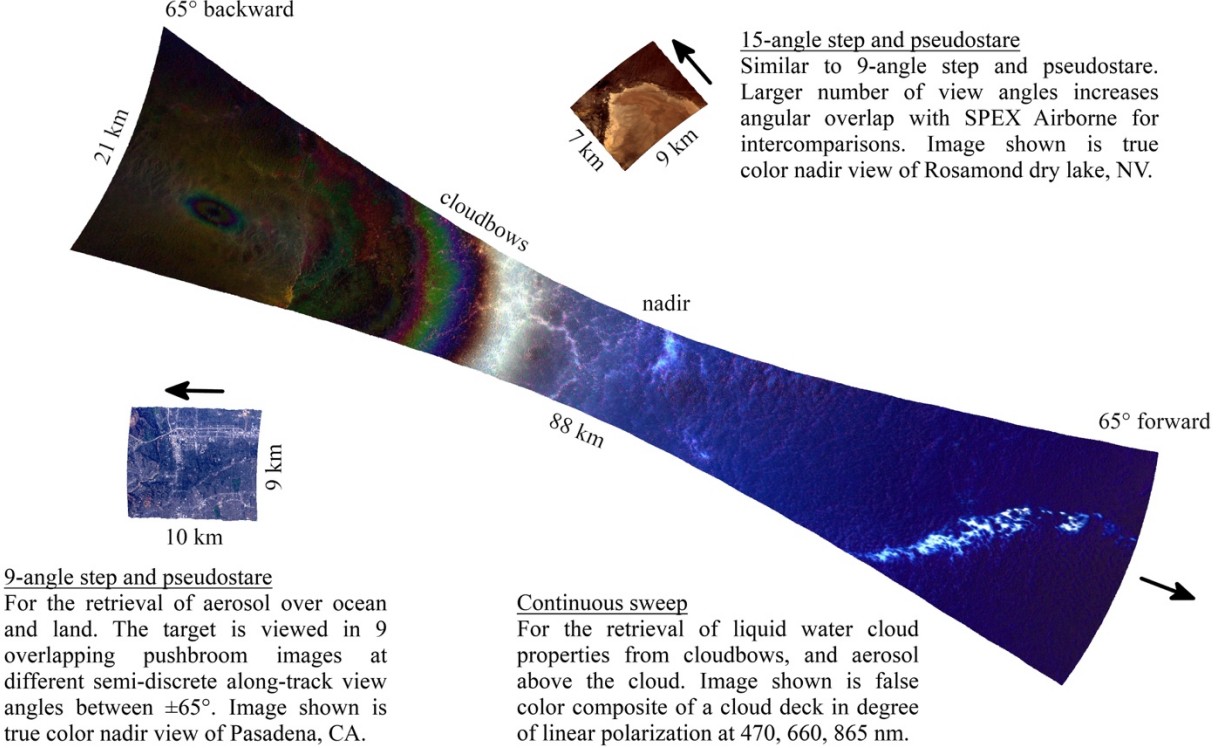

**15-angle step and pseudostare**
Similar to 9-angle step and pseudostare. Larger number of view angles increases angular overlap with SPEX Airborne for intercomparisons. Image shown is true color nadir view of Rosamond dry lake, NV.

**9-angle step and pseudostare**
For the retrieval of aerosol over ocean and land. The target is viewed in 9 overlapping pushbroom images at different semi-discrete along-track view angles between ±65°. Image shown is true color nadir view of Pasadena, CA.

**Continuous sweep**
For the retrieval of liquid water cloud properties from cloudbows, and aerosol above the cloud. Image shown is false color composite of a cloud deck in degree of linear polarization at 470, 660, 865 nm.

**Figure 5 AirMSPI quicklooks for the 3 different multi-angle observation modes. Aircraft flight direction is indicated by arrows.**


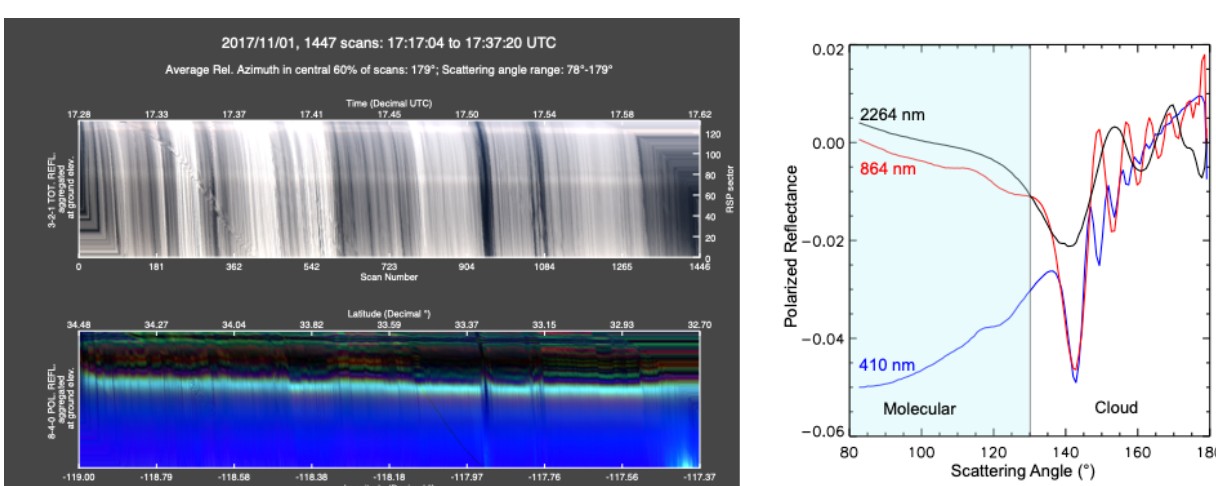

**Figure 6 RSP quick look image (left) from November 1st, 2017. The vertical dimension represents view angle, and the horizontal dimension scan number (time elapsed. At right is a single level 1C scan corresponding to scan 1235 at left. It shows the signature of cloud bows at large scattering angles (135-180˚) and molecular (Rayleigh) scattering for 410 nm at scattering angles smaller than 130°. Retrieved droplet size parameters are effective radius of 11.5 μm and effective variance of 0.008.**


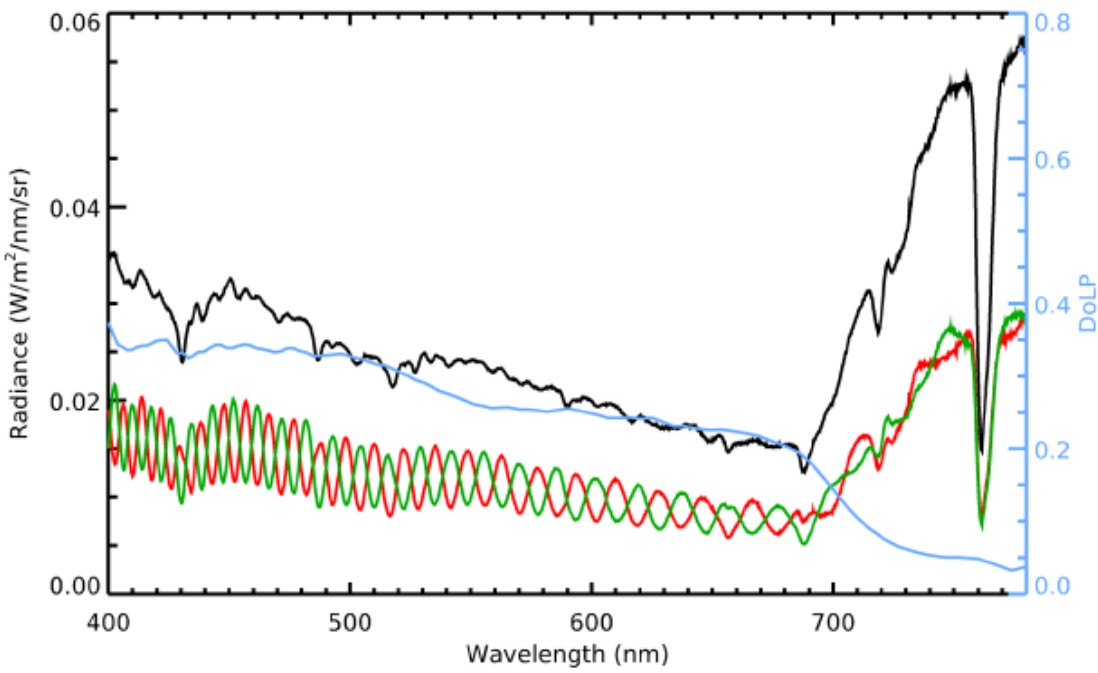

**Figure 7 Example of SPEX airborne measurements over vegetation, for radiance (black line) and DoLP (blue line) as function of wavelength. The modulated spectra from which radiance and DoLP are derived are shown in green and red.**

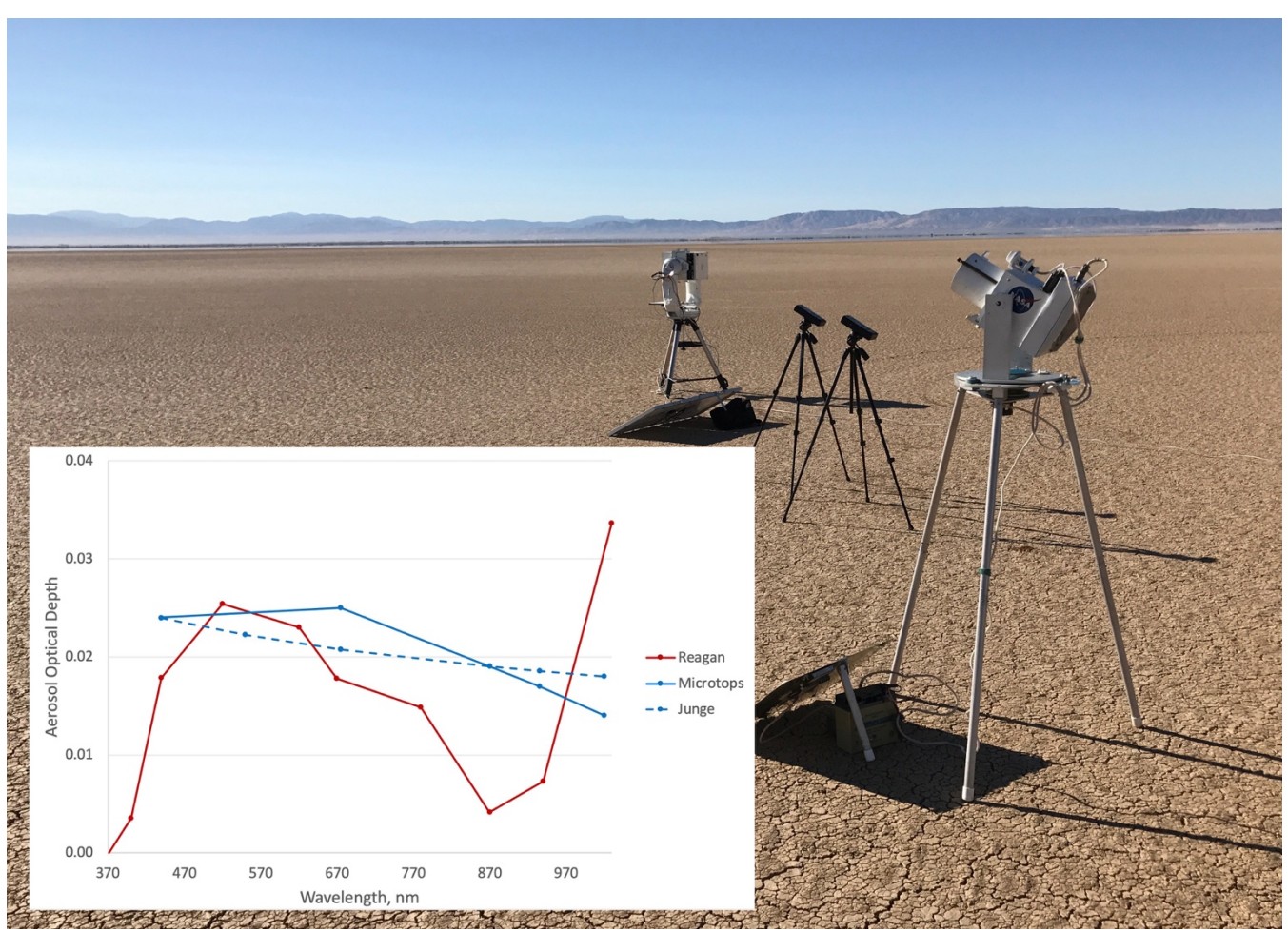

**Figure 8 GroundMSPI, two Microtops sunphotometers, and a University of Arizona Reagan sunphotometer deployed on October 25th, 2017 at Rosamond Dry Lake in California. (Inset) AOD as measured by the Microtops and Reagan sunphotometers, plus a Junge model fit to the Microtops data.**

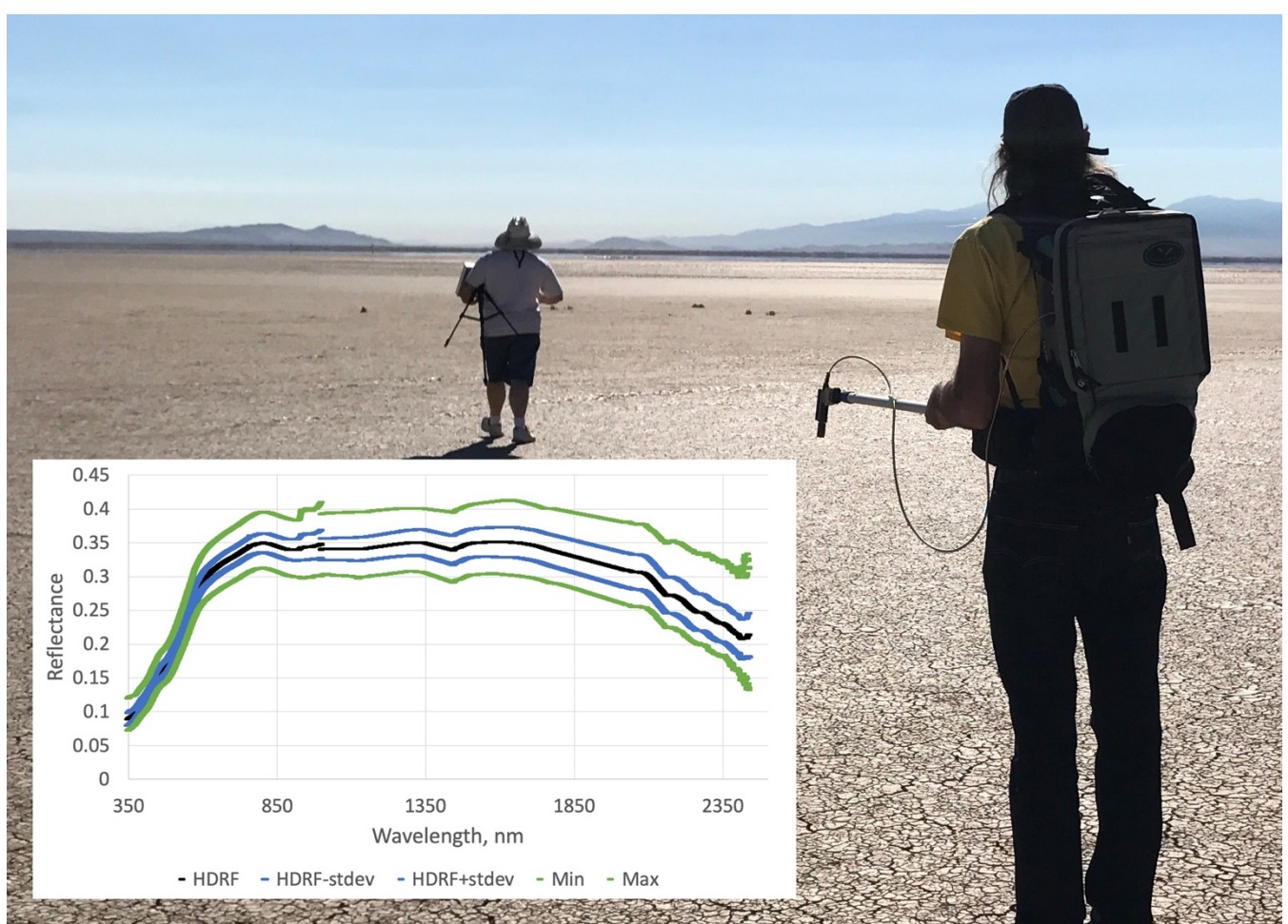

**Figure 9 Surface characterization on October 25th, 2017 at Rosamond Dry Lake in California using a Spectralon reflectance standard and ASD FieldSpecPro. (Inset) Mean and standard deviation of surface reflectance (black, blue lines) with maxima and minima spectra (green). Reflectances are presented in terms of the Hemispherical-Directional Reflectance Factor (HDRF), Schaepman-Strub et al., 2006.**

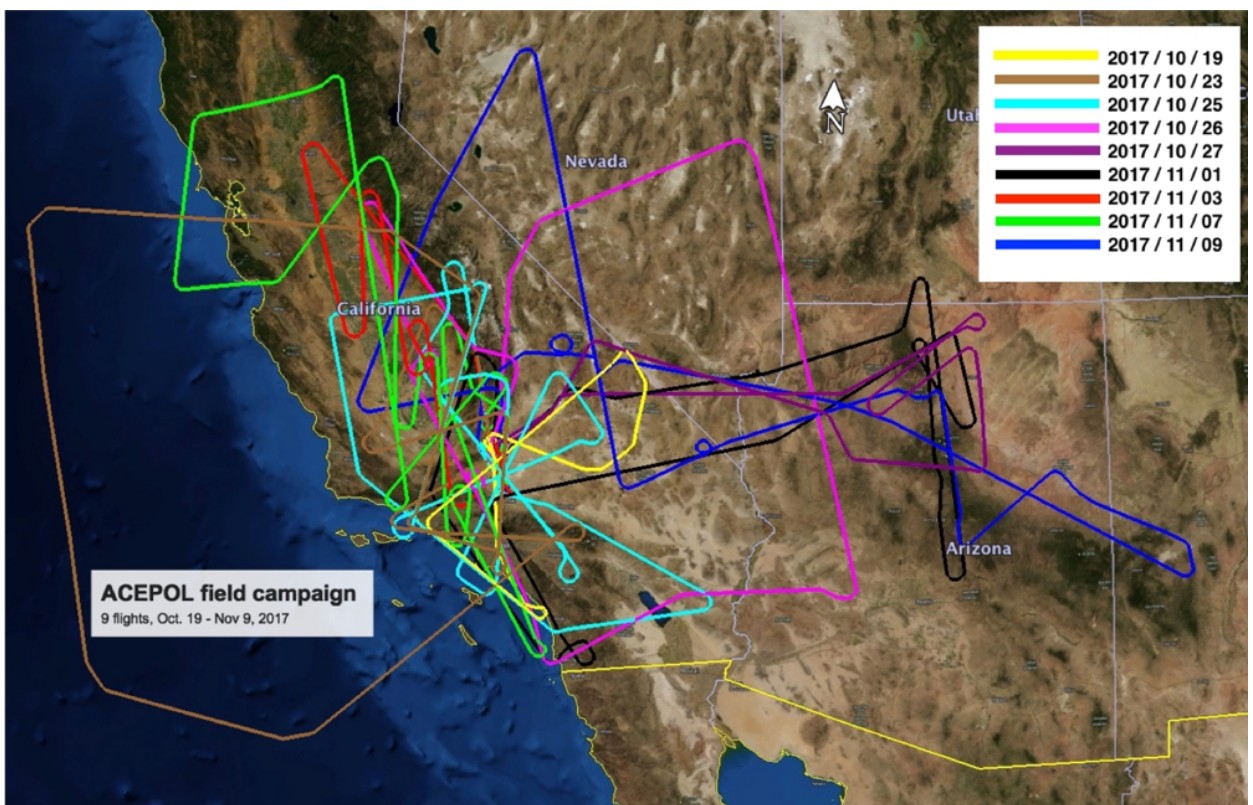

**Figure 10 Flight tracks for the ACEPOL field campaign. Nine flights were conducted from October 19th to November 9th, 2017 over California, Nevada, Arizona, New Mexico and the coastal Pacific Ocean from the Armstrong Flight Research Center in Palmdale, California. Image mapped using Google Earth, © 2018 Google.**

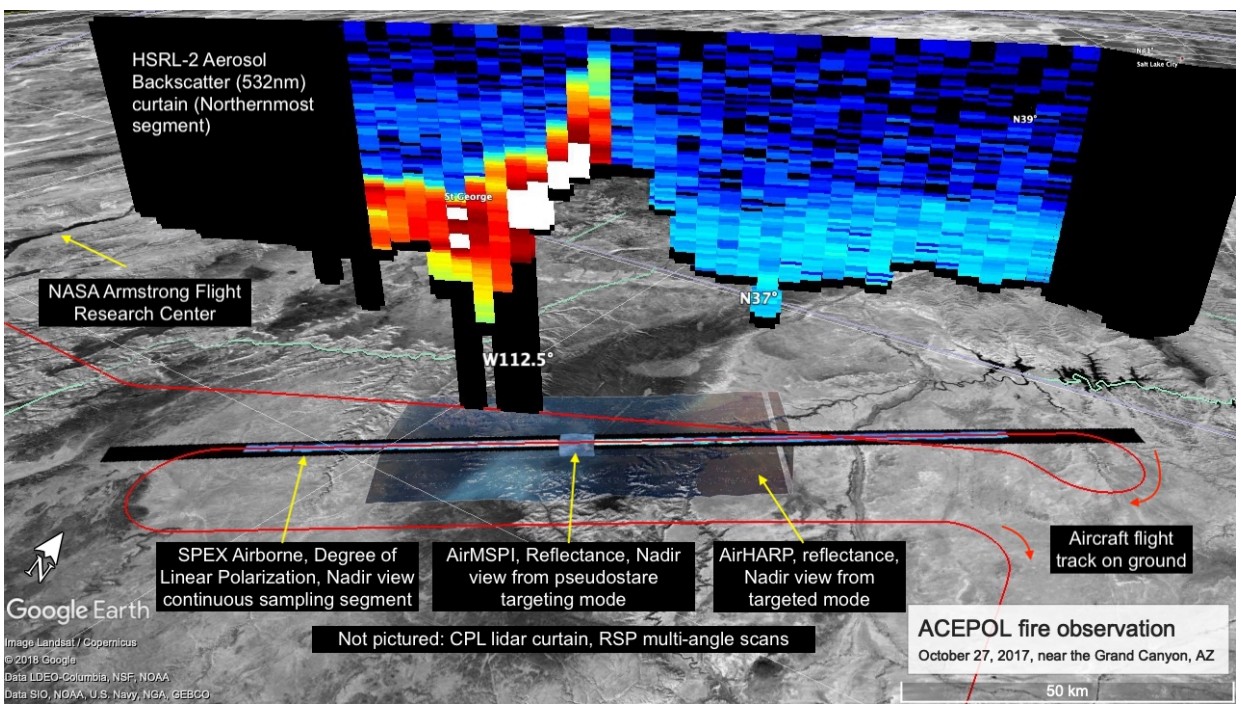

**Figure 11 Example of variable instrument observation characteristics for a scene on October 27, 2017, of biomass burning smoke from prescribed burns near the Grand Canyon. Image mapped using Google Earth, © 2018 Google.**