# Peer review of "The Aerosol Characterization from Polarimeter and Lidar (ACEPOL) airborne field campaign"

_Earth System Science Data, 2020_

## Referee Comment (RC1) · Meloe Kacenelenbogen (Referee) · 21 May 2020

This paper is an overview of the ACEPOL airborne field campaign. Its purpose is to document ACEPOL's initial hypothesis, science objectives, atmospheric conditions, field deployment, airborne and ground-based instruments, and scientific outcomes. It is, for the most part, well structured, clear, and well-written. I consider the necessary revisions minor as they do not involve any extensive data analysis. However, others might consider them major as they are beyond the correction of a few typos. Overview publications of airborne field campaigns are unique and, in my opinion, very useful. I recommend this manuscript for publication only after the following comments and

suggestions are addressed.

Overall comments . Data archive and availability should ideally be described in one place instead of many (e.g., in each instrument sections 3.1.1- 3.1.4 and section 3.4) . Consider adding a table of ACEPOL (and possibly PODEX) related papers (e.g., published, in review, in prep i.e. conferences or technical reports), instruments involved and the science objectives they mainly address . Section 2 (Objectives) should state and describe ACEPOL's objectives more clearly and describe how ACEPOL has addressed each one of them. . Consider explaining more clearly how ACEPOL differs from or complements PODEX . Consider describing ACEPOL vicarious calibration in more details and which instruments are concerned . Some figures need further description and analysis, especially the ones showing scientific results such as spectral AOD comparison, reflectance (e.g., Fig. 4, 8, 9) . As an illustration, consider showing Aerosol Optical Depths derived from AirMSPI, RSP and SPEX (in addition to AOD from AirHARP on Fig. 4)

Detailed comments Line 34 (and any other place in the manuscript): It should say HSRL-2 instead of HSRL2. Line 47: "While existing passive . . . (Mishchenko et al., 2004)." This statement could use more information. Line 55: "previous ocean color satellite instruments" could use some examples Line 74: CATS needs a reference, possibly McGill et al. [2015] McGill et al., 2015; McGill, M. J., J. E. Yorks, V. S. Scott, A. W. Kupchock, and P. A. Selmer (2015), The Cloud‐Aerosol Transport System (CATS): A technology demonstration on the International Space Station, Proc. SPIE 9612, Lidar Remote Sensing for Environmental Monitoring XV, 96120A, doi:10.1117/12.2190841. Line 87: replace "at" by "the" in "high altitude vantage point of at aircraft." Line 87: consider replacing "perhaps" by "to our knowledge" Line 99: objectives are (i) test new observations systems, (ii) develop new algorithms, or (iii) validate orbital observations. Consider giving a subsection to each of these three objectives and provide more information. For example, current section 2.2. would fall under objective (iii). Section 2.1 describes how ACEPOL relates to ACE and PODEX.

Consider including this in the introduction instead (giving it its own section outside of section 2). Line 101: "For that reason, a wide range of observation conditions were desired". Consider including examples of easy or challenging atmospheric conditions for specific MAPs or Lidars retrievals. Line 103: How were targets prioritized? Line 106: If the objectives are as general as "(i) test new observations systems, (ii) develop new algorithms, or (iii) validate orbital observations" (line 99), then BRDF, BPDF and ocean color algo. might still fall under (ii) and (iii)? Line 121: Consider listing MAPs and lidar(s) during PODEX. Were ACEPOL and PODEX designed to address the same science objectives? Were synergistic algorithm(s) involving MAPs and lidars not possible during PODEX? Why? Line 132: Can the authors add any publication, even in preparation, that will use ACEPOL data for a CALIOP/ CATS validation effort? Line 133: Does SRON have different objectives than the overarching ones on Line 99 (albeit more specific to SPEX)? Is there a reason to single out SRON in the objective section? Section 2.3 seems redundant with section 5.4 (and possibly 5.5) Line 137: NASA PACE mission needs a description as soon as introduced (instead of on line 164) Line 146: Is there no published reference for AirHARP? Line 161: HARP CubeSat instrument needs a reference too, possibly Vanderlei et al. [2016]. J. Vanderlei Martins, "HARP: Hyper-Angular Rainbow Polarimeter CubeSat," ESTO Science and Technology Forum, June 14, 2016, URL: https://esto.nasa.gov/forum/estf2016/PRESENTATIONS/Martins_A1P2_ESTF2016.pdf "HARP Hyper-Angular Rainbow Polarimeter," USU/SDL, URL: http://www.sdl.usu.edu/downloads/harp.pdf Does it have a scheduled launch date? Consider adding a reference to section 5.2 here. Line 163: "will provide global coverage two days," I don't understand this statement. Line 170: AirHARP AOD seems to reach 1 at 440 nm on Fig. 4d. Can you describe this briefly and possibly how it compares to other MAPs? Line 172: Here "McBride et al., 2019" instead of "McBride et al., 2019 in preparation" on line 158. Line 174: consider discussing data availability for AirHARP? In Fu et al., 2020, it says "Note that aerosol retrievals from AirHARP measurements are not included in this paper because the data were not available when

performing the analysis presented here" Line 184: "see section 5.3" instead of section 6.3 Line 196: change to Fu et al. [2020] Fu, Guangliang, et al. "Aerosol retrievals from different polarimeters during the ACEPOL campaign using a common retrieval algorithm." (2020). Line 229: consider referring to section 3.4 for data availability and to Fu et al. [2020] (and any other publications I am missing) concerning RSP-ACEPOL specific analysis Line 248: "Fu et al., 2020" Line 279: "MuÌLller et al., Sawamura et al." need years. Another good reference for HSRL-2 might be Burton et al. [2018] Burton, S. P., et al. "Calibration of a high spectral resolution lidar using a Michelson interferometer, with data examples from ORACLES." Applied optics 57.21 (2018): 6061-6075. Line 281: correct to "extinction" instead of "depolarization" at 355nm (i.e., 3 beta + 2 alpha and depolarization at 3 wavelengths for HSRL-2 instead of 2 beta + 1 alpha and depolarization at 2 wavelengths for HSRL-1) Line 313: consider replacing "inversion products" by aerosol intensive properties (single scattering albedo, size distribution etc.). Line 322: should say aerodynamic diameter Line 324: should say "gaseous criteria for . . ."? Line 334: consider describing "ACEPOL vicarious calibration efforts" and which remote sensing instrument this concerns. Line 339: consider adding a reference for "Reagan sunphotometer", possibly Bruegge et al., [1990] that describes this sunphotometer. This paper also references Shaw et al. [1973] Bruegge, Carol J., et al. "In-situ atmospheric water-vapor retrieval in support of AVIRIS validation." Imaging spectroscopy of the terrestrial environment. Vol. 1298. International Society for Optics and Photonics, 1990. Shaw, G.E., Reagan, J. A., and Herman, B. M. (1973), Investigations of atmospheric extinction using direct solar radiation measurements made with a multiple wavelength radiometer, J. Appi. Meteorol. 12:374-380. Line 349: The difference between Reagan and microtops AOD at specific wavelengths (Fig. 8) should be explained in the text Line 386: correct "Figure 10 Flight tracks for the ACEPOL field campaign.is a graphical illustration of ACEPOL flight tracks.". Also, "Details on the characteristics of each flight are in Table 4" seems to be a repeat of line 374 Line 388: consider explaining "somewhat unusual conditions" Line 391: "constrain observations by": consider specifying which airborne remote sensing instrument (or

algorithm) needs to be constrained Line 415: "test of retrieval capability". Consider briefly reporting the results from Fu et al. [2020] or any other relevant publication as an example Line 425: consider more specifics on how ACEPOL data are used in the ACCP efforts (e.g., used to constrain canonical cases in theoretical studies or to attribute instrument specific uncertainties or to attribute scores to different candidate satellite architectures in respect to specific science objectives?) Line 441: consider adding any studies underway (i.e., studies that use ACEPOL-AirHARP to help HARP CubeSat design and algorithm) Line 445: MAIA will provide speciated PMs (said on Line 457). This is one step further from aerosol "typing". Line 454: delete "total and" in "retrieve total and AOD" Line 457: "A geostatistical regression modeling framework will be used to transform column aerosol optical and microphysical properties to speciated, near-surface PM concentrations". Consider possibly adding Kalashnikova et al. [2018] here and any other (more) relevant publication Kalashnikova, O. V., Garay, M. J., Bates, K. H., Kenseth, C. M., Kong, W., Cappa, C. D., et al. (2018). Photopolarimetric sensitivity to black carbon content of wildfire smoke: Results from the 2016 ImPACT-PM field campaign. Journal of Geophysical Research: Atmospheres, 123. https://doi.org/10.1029/2017JD028032 Line 459: consider adding any studies underway (i.e., studies that use ACEPOL-AirMSPI to help MAIA design and algorithm) Line 472: Fu et al. [2020] Line 475: is this still happening in May 2020 (under COVID-19)? Line 476: MAMAP and NanoCarb need to be introduced and described Line 478: consider describing how ACEPOL-related airSPEX data will help those upcoming deployments as Section 5 is about "value (of ACEPOL) for future missions" Conclusion: please consider re-stating the ACEPOL objectives and how they were addressed, as well as recapping the value of ACEPOL in future missions (e.g., PACE, MAIA, HARP CubeSat, ACCP) Figure 4d: "tau" should be "AOD" next to the color bar. Figure 8: What does "Junge" stand for? Also the difference between Reagan and microtops AOD at specific wavelengths should be explained in the text Figure 9: What does "HDRF" stand for?

---

## Referee Comment (RC2) · Anonymous Referee #2 · 28 May 2020

General Comments This paper describes the ACEPOL campaign in order to test six observation instruments and new developed algorithms for accurate measurements of cloud and aerosol optical properties. The flight campaign and the data provided are useful for reducing aerosol-cloud climate forcing uncertainty. I believe this paper is suitable for publication to Earth System Science Data after considering comments as below.

Specific comments 1. The data provided by the ACEPOL campaign are important for this paper, so it's better to describe the specific data types, such as aerosol AOD, or vertical profile of cloud, in Table 3.

[Figure]

2. The five instruments and the corresponding data was calibrated except AirMSPI, does this instrument need radiometric calibration?

3. Section "Conclusions" should summarize more information about the data archived, including specific data type, spatial resolution, temporal resolution, etc.

Technical corrections Line 87 The sentence is difficult to be understood: "It is, perhaps, the closest an airborne instrument can get to a space deployment." Please rewrite this sentence.

Line 146 "... designed characterization of "-> "designed for characterization of"

Line 164 "mission due to launch in 2022"->" mission due to be launched in 2022"

Line 386 Remove "Flight tracks for the ACEPOL field campaign." from the sentence "Figure 10 Flight tracks for the ACEPOL field campaign.is a graphical illustration of ACEPOL flight tracks."

Reference Please make the format consistent for all references. For example, some references have DOI link, while some have no DOI.

Line 826 "angle produce a full pushbroom image"->"angle produces a full pushbroom image"

Figure 4(d) Lacks label for y-axis, i.e. "Latitude"

---

## Referee Comment (RC3) · Anonymous Referee #3 · 10 Jun 2020

Review of "The Aerosol Characterization from Polarimeter and Lidar (ACEPOL) airborne field campaign" by Knobelspiesse et al.

General:

This manuscript documents collection of atmospheric aerosols using 4 polarimeters and 2 lidars onboard ER-2 during ACEPOL. The CPL lidar has a horizontal resolution of about 200m and a vertical resolution of 30m, while the HSRL_2 has a horizontal resolution of 1-2 km, and a vertical resolution of 15m.

The 4 polarimeters onboard ER-2 contains spatial sampling from about 42.9 km, 9 km, 280 m per pixel, to 6-degree cross track. The resolutions range from about 10 m at ground to 1 km.

There were 9 ACEPOL flights conducted in October-November 2017. Dates of calibration of ACEPOL data were described in Table 2. There are 9 days with calibration days specified: 10/19, 10/23, 10/25, 10/26, 10/27, 11/01, 11/03, 11/07, 11/09.

Test of data accessibility (see below) indicates that the data is not user friendly. Please refer to the NASA GTE project, which contains lots of flights of data. The GTE data are very easy to use and user friendly.

Comments:

1. Calibrations: The calibrations effect, before and after the calibration of the observational data, is not clear. The details of flights are outlined in Table 4. Please provide results (in figures, etc) comparing raw data with the calibrated data.
2. What are the variables observed during the 9 flights? What are the time resolution of data of each observed variable?
3. Test of data accessibility from Table 3:

ASDC:   https://eosweb.larc.nasa.gov/10.5067/SUBORBITAL/ACEPOL2017/DATA001

[Figure]

AirMSPI

https://eosweb.larc.nasa.gov/project/airmspi/airmspi_table/10.5067/AIRCRAFT/AIRMSPI/ACEPOL/RADIANCE/ELLIPSOID_V006

https://eosweb.larc.nasa.gov/project/airmspi/airmspi_table/10.5067/AIRCRAFT/AIRMSPI/ACEPOL/RADIANCE/TERRAIN_V006

[Figure]

GroundMSPI :
https://eosweb.larc.nasa.gov/project/airmspi/airmspi_table/10.5067/GROUND/GROUNDMSPI/ACEPOL/RADIANCE_v009

[Figure]

AERONET https://aeronet.gsfc.nasa.gov/

[Figure]

Q: Where to find data relevant to the calibration of the 9 ER-2 flights described in Table 2?

CARB https://www.arb.ca.gov/adam/index.html

[Figure]

Q: Where to get data relevant to 9 ER-2 flights in this work from this page?

4. Test of data accessibility from Abstract and reference ACEPOL Science Team, 2017:
   doi:10.5067/SUBORBITAL/ACEPOL2017/DATA001
   Test results: I was not able to find data from input above line to google.

   But I was managed to find data from following link:
   https://asdc.larc.nasa.gov/project/ACEPOL/ACEPOL_AircraftRemoteSensing_CPL_Data_1

[Figure]

   After trying to Get Dataset, with OPENDATA selected in Additional Options, following page poped up:

[Figure]

Even in here, it is still unclear where and how to view the flight data?

---

## Author Comment (AC1) · 17 Jul 2020

**Response to reviewers of "The Aerosol Characterization from Polarimeter and Lidar (ACEPOL) airborne field campaign," Knobelspiesse et al, ESSD-2020-76**

https://doi.org/10.5194/essd-2020-76

**Reviewer #1 (Meloe Kacenelenbogen)**

This paper is an overview of the ACEPOL airborne field campaign. Its purpose is to document ACEPOL's initial hypothesis, science objectives, atmospheric conditions, field deployment, airborne and ground-based instruments, and scientific outcomes. It is, for the most part, well structured, clear, and well-written. I consider the necessary revisions minor as they do not involve any extensive data analysis. However, others might consider them major as they are beyond the correction of a few typos. Overview publications of airborne field campaigns are unique and, in my opinion, very useful. I recommend this manuscript for publication only after the following comments and suggestions are addressed.

*We thank the reviewer for her thoughtful and through comments. They improve the manuscript and we are grateful for the time she spent reviewing our manuscript. We believe we can accommodate or respond to all comments.*

**Overall comments.**

Data archive and availability should ideally be described in one place instead of many (e.g., in each instrument sections 3.1.1- 3.1.4 and section 3.4).

*Section 3.4 ("Data Availability") and Table 3 (in the same section) are the primary location describing archive location and availability. Additional details in the abstract are a requirement of the ESSD journal format. Archive details in each instrument section are not presented, instead those sections point to section 3.4 for details on this.*

Consider adding a table of ACEPOL (and possibly PODEX) related papers (e.g., published, in review, in prep i.e. conferences or technical reports), instruments involved and the science objectives they mainly address.

*While this is a good idea, we're afraid it would become rapidly out of date since research with ACEPOL data is ongoing. Instead, we are looking into creating such a table on the primary archive landing page, which can be updated as new publications are created.*

Section 2 (Objectives) should state and describe ACEPOL's objectives more clearly and describe how ACEPOL has addressed each one of them.

*To accomplish this, we added the following two paragraphs to section 2. More details of the achievements for specific objectives follow.*

*"Target types fell into four broad categories: calibration, geolocation, validation, and targets of opportunity. Calibration targets (1a, 1b, and 1c) were meant to provide spatially uniform observations with which radiometric and polarimetric measurements between multi-angle polarimeters can be compared. A similar intercomparison was performed during the Polarimeter Definition Experiment (PODEX) between the RSP and AirMSPI instruments (Knobelspiesse et al., 2019). Intercomparison can now be performed with those instruments plus AirHARP and SPEX Airborne. Such intercomparisons can confirm measurement uncertainty estimates and identify calibration problems. Different scene types are useful for intercomparison: cloud free ocean observations (1a) provide low reflectance, potentially highly polarized measurements, while land scenes can have high reflectance but moderate to low polarization. Cloud scenes provide high reflectance and low polarization. Intercomparison requires accurate geolocation, so minimizing scene heterogeneity and atmospheric variation is important. Furthermore, scenes with distinct features, such as coastlines, were targeted to provide a geolocation reference (1d). Validation targets are used to test the geophysical products retrieved by the airborne sensors against similar observations on the ground (2a, 2b), by other field campaigns (2c), and by satellites (3a, 3b). Targets of opportunity are intended for algorithm development and represent infrequently observed or difficult scenes. Finally, it should be noted that the target number designation roughly indicates priority. Low numbered targets are of highest priority, and are generally organized such that targets supporting validation or calibration of radiometric quantities have greatest precedence, followed by validation of geophysical products derived from such observations, and then special cases and difficult scenes (targets of opportunity).*

*Most of the highest priority targets were observed successfully during ACEPOL, although conditions precluded observation of uniform marine stratocumulus cloud decks (target 1c), and most cases of high aerosol loads. The latter was highly unusual for this part of the world, as California's San Joaquin Valley and the Los Angeles metropolitan area are known for typically high aerosol loads. The solution was to overfly controlled forest fire burns farther afield in Arizona. Attempted coordination with the Coupled Air Sea Processes and EM Ducting Research (CASPER) East (Wang et al, 2018) field campaign was unfortunately not possible because of scheduling difficulty and weather. Serendipitously, one flight overlapped with a flight by the Alpha Jet Atmospheric Experiment (AJAX), which carried a payload of atmospheric gas sensors. High aerosol loads over the ocean (4a) were not observed, but low aerosol load overflights of an AERONET site (2b) on a platform off Long Beach, CA have become the basis for several analysis papers (e.g. Fu et al, 2019, Gao et al, 2020). Another important accomplishment was the successful overflight of Rosamond Dry Lake from multiple headings while a ground based team characterized the spectral reflectance of the lakebed. This was used to*

*vicariously adjust the AirMSPI calibration, and serves as a reference for other measurements as well."*

Consider explaining more clearly how ACEPOL differs from or complements PODEX.

*More detail was added to section 2.1.*

Consider describing ACEPOL vicarious calibration in more details and which instruments are concerned

*We expanded on this in section 3.3.3*

Some figures need further description and analysis, especially the ones showing scientific results such as spectral AOD comparison, reflectance (e.g., Fig. 4, 8, 9) As an illustration, consider showing Aerosol Optical Depths derived from AirMSPI, RSP and SPEX (in addition to AOD from AirHARP on Fig. 4)

*Our general philosophy in this manuscript has been to illustrate the types of data comparisons that can be made, and provide the tools for interested members of the community to identify data most appropriate for their needs. In that light, Figures 4, 8, 9 are meant to be illustrative, and scientific results are the domain of cited and future publications. This delineation corresponds to the scope of the ESSD journal, which states: "Any interpretation of data is outside the scope of regular articles."*

**Detailed comments**

Line 34 (and any other place in the manuscript): It should say HSRL-2 instead of HSRL2.
*Corrected, thank you*

Line 47: "While existing passive . . . (Mishchenko et al., 2004)." This statement could use more information.
*Added the sentence: "In other words, the remote sensing retrieval solutions are often non-unique, and require the use of constraints in the form of, for example, aerosol models, which may or may not represent geophysical reality."*

Line 55: "previous ocean color satellite instruments" could use some examples
*Reference to SeaWiFS added*

Line 74: CATS needs a reference, possibly McGill et al. [2015] McGill et al., 2015; McGill, M. J., J. E. Yorks, V. S. Scott, A. W. Kupchock, and P. A. Selmer (2015), The Cloud‐Aˆ R˘ Aerosol Transport System (CATS): A technology demonstration on the International Space Station, Proc. SPIE 9612, Lidar Remote Sensing for Environmental Monitoring XV, 96120A, doi:10.1117/12.2190841.

*A good suggestion, thank you*

Line 87: replace "at" by "the" in "high altitude vantage point of at aircraft."

*Done, thanks*

Line 87: consider replacing "perhaps" by "to our knowledge"

*We modified this sentence to: "Deployment of an instrument on this aircraft is a close analog for the space environment and observation conditions."*

Line 99: objectives are (i) test new observations systems, (ii) develop new algorithms, or (iii) validate orbital observations. Consider giving a subsection to each of these three objectives and provide more information. For example, current section 2.2. would fall under objective (iii).

*We expanded on this considerably as noted above.*

Section 2.1 describes how ACEPOL relates to ACE and PODEX. Consider including this in the introduction instead (giving it its own section outside of section 2).

*ACE was one of three providers of funding supporting ACEPOL, so we would like to keep this in a separate, sub-level section (2.1 vs 2, when CALIPSO and SRON support is described in 2.2 and 2.3). However, we did add some text to section 2.1 to better describe the relationship between ACEPOL and PODEX, as suggested above.*

Line 101: "For that reason, a wide range of observation conditions were desired". Consider including examples of easy or challenging atmospheric conditions for specific MAPs or Lidars retrievals.

*We put in some language that "targets of opportunity"*

Line 103: How were targets prioritized?
*We put some test in section 2 on this:*

> *Low numbered targets are of highest priority and are generally organized such that targets supporting validation or calibration of radiometric quantities have greatest precedence, followed by validation of geophysical products derived from such observations, and then special cases and difficult scenes (targets of opportunity).*

Line 106: If the objectives are as general as "(i) test new observations systems, (ii) develop new algorithms, or (iii) validate orbital observations" (line 99), then BRDF, BPDF and ocean color algo. might still fall under (ii) and (iii)?

*Perhaps correct. But we'd like to include this wording to indicate that these less obvious objectives might be achievable, and even by other instruments (we added that note).*

Line 121: Consider listing MAPs and lidar(s) during PODEX. Were ACEPOL and PODEX designed to address the same science objectives? Were synergistic algorithm(s) involving MAPs and lidars not possible during PODEX? Why?

*Section 2.1 has been reworded such that it now lists the MAPs and Lidars used during ACEPOL (which were RSP, AirMSP, PACS and CPL. PACS didn't work). While there is some overlap between PODEX and ACEPOL objectives, the latter was important because several new instrument (concepts) were available: AirHARP, SPEX Airborne and HSRL-2.*

Line 132: Can the authors add any publication, even in preparation, that will use ACEPOL data for a CALIOP/ CATS validation effort?

*Thanks, some references for this were added.*

Line 133: Does SRON have different objectives than the overarching ones on Line 99 (albeit more specific to SPEX)? Is there a reason to single out SRON in the objective section? Section 2.3 seems redundant with section 5.4 (and possibly 5.5)

*Although the technical objectives of ACE and SRON have significant overlap, we need to show that actual funding was supplied by SRON to support ACEPOL flights. SRON's objectives are the development of the SPEX concept, which is participating in PACE, but is under consideration for other missions as well.*

Line 137: NASA PACE mission needs a description as soon as introduced (instead of on line 164)

*A good point, we added more detail and a description of PACE to section 2.3.*

Line 146: Is there no published reference for AirHARP?

*AirHARP characteristics are best described in the McBride and Puthukkudy papers, which are cited.*

Line 161: HARP CubeSat instrument needs a reference too, possibly Vanderlei et al. [2016]. J. Vanderlei Martins, "HARP: Hyper-Angular Rainbow Polarimeter CubeSat," ESTO Science and Technology Forum, June 14, 2016, URL: https://esto.nasa.gov/forum/estf2016/PRESENTATIONS/Martins_A1P2_ESTF2016.pdf "HARP Hyper-Angular Rainbow Polarimeter," USU/SDL, URL: http://www.sdl.usu.edu/downloads/harp.pdf Does it have a scheduled launch date? Consider adding a reference to section 5.2 here.

*HARP CubeSat has been launched and is producing data!*

Line 163: "will provide global coverage two days," I don't understand this statement.
*Missing a crucial 'in'. Given orbit and swath of PACE/HARP2, all points on the globe will be imaged in the course of two days.*

Line 170: AirHARP AOD seems to reach 1 at 440 nm on Fig. 4d. Can you describe this briefly and possibly how it compares to other MAPs?

*The large aerosol optical depth in that scene is unsurprising given that this is a forest fire smoke plume at its source. Puthukkudy et. al 2020 has a comparison of HSRL-2 measured AOD with AirHARP retrieved AOD at 532 nm for the same smoke scene. The results show that there is good agreement with HSRL-2 measured AOD with exception of some points near to the smoke source. The spatial resolution mismatch between HSRL-2 and AirHARP measurements are the main reason for this. For the same smoke scene, Fu et. al 2020 also shows that there is a good agreement between the retrieved AOD from SPEX airborne, and RSP with the HSRL-2 measurements.*

Line 172: Here "McBride et al., 2019" instead of "McBride et al., 2019 in preparation" on line 158.
*This has been updated.*

Line 174: consider discussing data availability for AirHARP? In Fu et al., 2020, it says "Note that aerosol retrievals from AirHARP measurements are not included in this paper because the data were not available when performing the analysis presented here"

*Currently, most AirHARP data are in the archive. There was an initial delay as calibration techniques were resolved.*

Line 184: "see section 5.3" instead of section 6.3

*thanks*

Line 196: change to Fu et al. [2020] Fu, Guangliang, et al. "Aerosol retrievals from different polarimeters during the ACEPOL campaign using a common retrieval algorithm." (2020).

*Updated, thanks. They changed the title from discussion to final version of this paper.*

Line 229: consider referring to section 3.4 for data availability and to Fu et al. [2020] (and any other publications I am missing) concerning RSP-ACEPOL specific analysis

*Reference added*

Line 248: "Fu et al., 2020" Line 279: "Muĺ Lller et al., Sawamura et al." need years. Another good reference for HSRL-2 might be Burton et al. [2018] Burton, S. P., et al. "Calibration of a high spectral resolution lidar using a Michelson interferom- eter, with data examples from ORACLES." Applied optics 57.21 (2018): 6061-6075.

*Thanks, the references were updated as noted and now represent how the HSRL team desires to report citations.*

Line 281: correct to "extinction" instead of "depolarization" at 355nm (i.e., 3 beta + 2 alpha and depolarization at 3 wavelengths for HSRL-2 instead of 2 beta + 1 alpha and depolarization at 2 wavelengths for HSRL-1)

*Thank you, the text is correct now.*

Line 313: consider replacing "inversion products" by aerosol intensive properties (single scattering albedo, size distribution etc.).

*Done, thanks*

Line 322: should say aerodynamic diameter

*Yes, thank you*

Line 324: should say "gaseous criteria for . . ."?

*Got it, thanks*

Line 334: consider describing "ACEPOL vicarious calibration efforts" and which remote sensing instrument this concerns.

*More details regarding this were added to that section.*

Line 339: consider adding a reference for "Reagan sunphotometer", possibly Bruegge et al., [1990] that describes this sunphotometer. This paper also references Shaw et al. [1973] Bruegge, Carol J., et al. "In-situ atmospheric water-vapor retrieval in support of AVIRIS validation." Imaging spectroscopy of the terrestrial environment. Vol. 1298. International Society for Optics and Photonics, 1990. Shaw, G.E., Reagan, J. A., and Herman, B. M. (1973), Investigations of atmospheric extinction using direct solar radiation measurements made with a multiple wavelength radiometer, J. Appi. Meteorol. 12:374-380.

*More details and references were added.*

Line 349: The difference between Reagan and microtops AOD at specific wavelengths (Fig. 8) should be explained in the text

*This is described now in the text – there is possibly a pressure error for the Microtops instrument, so the Reagan was used as a reference.*

Line 386: correct "Figure 10 Flight tracks for the ACEPOL field campaign.is a graphical illustration of ACEPOL flight tracks.". Also, "Details on the characteristics of each flight are in Table 4" seems to be a repeat of line 374

*Fixed.*

Line 388: consider explaining "somewhat unusual conditions"

*Added 'low aerosol'*

Line 391: "constrain observations by": consider specifying which airborne remote sensing instrument (or algorithm) needs to be constrained

*Reworded to say "validate retrievals by"*

Line 415: "test of retrieval capability". Consider briefly reporting the results from Fu et al. [2020] or any other relevant publication as an example

*While it is probably out of scope to go into too much detail, we added reference to a number of publications here that analyze ACEPOL data*

Line 425: consider more specifics on how ACEPOL data are used in the ACCP efforts (e.g., used to constrain canonical cases in theoretical studies or to attribute instrument specific uncertainties or to attribute scores to different candidate satellite architectures in respect to specific science objectives?)

*This is an ongoing activity, but as far as we are aware is in it's preliminary stages. It has not yet been published in the literature.*

Line 441: consider adding any studies underway (i.e., studies that use ACEPOL-AirHARP to help HARP CubeSat design and algorithm)

*This section was updated*

Line 445: MAIA will provide speciated PMs (said on Line 457). This is one step further from aerosol "typing".

*Language updated*

Line 454: delete "total and" in "retrieve total and AOD"

*thanks*

Line 457: "A geostatistical regression modeling framework will be used to transform column aerosol optical and microphysical properties to speciated, near-surface PM concentrations". Consider possibly adding Kalashnikova et al. [2018] here and any other (more) relevant publication Kalashnikova, O. V., Garay, M. J., Bates, K. H., Kenseth, C. M., Kong, W., Cappa, C. D., et al. (2018). Photopolarimetric sensitivity to black carbon content of wildfire smoke: Results from the 2016 ImPACT-PM field campaign. Journal of Geophysical Research: Atmospheres, 123. https://doi.org/10.1029/2017JD028032

*We added this reference, thank you.*

Line 459: consider adding any studies underway (i.e., studies that use ACEPOL-AirMSPI to help MAIA design and algorithm)

*As far as we are aware, no such studies are underway, at least in the published literature.*

Line 472: Fu et al. [2020]

*Thanks*

Line 475: is this still happening in May 2020 (under COVID-19)?

*The SCARBO campaign is currently scheduled for September, 2020, depending on the COVID-19 situation. Text has been updated to reflect this.*

Line 476: MAMAP and NanoCarb need to be introduced and described

*More background information and citations are now provided.*

Line 478: consider describing how ACEPOL-related airSPEX data will help those upcoming deployments as Section 5 is about "value (of ACEPOL) for future missions"

*We added a sentence for this. In addition to previously mentioned work, the data processing system developed for ACEPOL is being used for future field campaigns.*

Conclusion: please consider re-stating the ACEPOL objectives and how they were addressed, as well as recapping the value of ACEPOL in future missions (e.g., PACE, MAIA, HARP CubeSat, ACCP)

*We added more description of this in the short conclusion.*

Figure 4d: "tau" should be "AOD" next to the color bar.
*done*

Figure 8: What does "Junge" stand for? Also the difference between Reagan and microtops AOD at specific wavelengths should be explained in the text

*This was a Junge aerosol model fit to the Microtops data. We updated the text to explain this.*

Figure 9: What does "HDRF" stand for?

*The Hemispherical-Directional Reflectance Factor. We spelled this out and provided the appropriate reference.*

**Reviewer #2**

**General Comments**

This paper describes the ACEPOL campaign in order to test six observation instruments and new developed algorithms for accurate measurements of cloud and aerosol optical properties. The flight campaign and the data provided are useful for reducing aerosol-cloud climate forcing uncertainty. I believe this paper is suitable for publication to Earth System Science Data after considering comments as below.

*We appreciate your sentiment and comments.*

Specific comments 1. The data provided by the ACEPOL campaign are important for this paper, so it's better to describe the specific data types, such as aerosol AOD, or vertical profile of cloud, in Table 3.

*Since the focus of ACEPOL is instrument and algorithm development, so called "Level 1" data are in the archive. Level 1 data are calibrated observables, such as radiance or reflectance, polarization, or backscatter. Level 2 data are products of retrieval algorithms and include, for example, aerosol intensive and extensive parameters. These may or may not exist in the database(s) depending on the retrieval algorithm maturity. We made a note of this in the table description.*

2. The five instruments and the corresponding data was calibrated except AirMSPI, does this instrument need radiometric calibration?

*All instruments were calibrated, albeit with different techniques and facilities. AirMSPI uses a vicarious adjustment to calibration coefficients based upon the ground based observations of the Rosamond Dry Lake.*

3. Section "Conclusions" should summarize more information about the data archived, including specific data type, spatial resolution, temporal resolution, etc.

*We updated the Conclusions to provide a more detailed description of the objectives of ACEPOL and the aspects of those objectives that were successfully retrieved. The resolutions, spectral character and other aspects of data type are described in Table 1.*

**Technical corrections**

Line 87 The sentence is difficult to be understood: "It is, perhaps, the closest an airborne instrument can get to a space deployment." Please rewrite this sentence.

*We modified to: Deployment of an instrument on this aircraft is a close analog for the space environment and observation conditions*

Line 146 ". . . designed characterization of "-> "designed for characterization of"

*Corrected, thank you*

Line 164 "mission due to launch in 2022"->" mission due to be launched in 2022"

*Considering the current pandemic, modified to "no earlier than 2022" as it is likely the launch date will slip.*

Line 386 Remove "Flight tracks for the ACEPOL field campaign." from the sentence "Figure 10 Flight tracks for the ACEPOL field campaign.is a graphical illustration of ACEPOL flight tracks."

*Corrected, thank you*

Reference Please make the format consistent for all references. For example, some references have DOI link, while some have no DOI.

*Ok*

Line 826 "angle produce a full pushbroom image"->"angle produces a full pushbroom image"

*Fixed, thanks*

Figure 4(d) Lacks label for y-axis, i.e. "Latitude"

*Done. Thank you.*

**Reviewer 3**

**General:**

This manuscript documents collection of atmospheric aerosols using 4 polarimeters and 2 lidars onboard ER-2 during ACEPOL. The CPL lidar has a horizontal resolution of about 200m and a vertical resolution of 30m, while the HSRL_2 has a horizontal resolution of 1-2 km, and a vertical resolution of 15m.

The 4 polarimeters onboard ER-2 contains spatial sampling from about 42.9 km, 9 km, 280 m per pixel, to 6-degree cross track. The resolutions range from about 10 m at ground to 1 km. There were 9 ACEPOL flights conducted in October-November 2017. Dates of calibration of ACEPOL data were described in Table 2. There are 9 days with calibration days specified: 10/19, 10/23, 10/25, 10/26, 10/27, 11/01, 11/03, 11/07, 11/09.

Test of data accessibility (see below) indicates that the data is not user friendly. Please refer to the NASA GTE project, which contains lots of flights of data. The GTE data are very easy to use and user friendly.

*We appreciate the attention of the reviewer. We presume you're referring to the Global Tropospheric Experiment series of field campaigns from the 1980's and 1990's. The ad hoc style of data archive website used for GTE is quite out of date and we presume difficult to maintain. Indeed, most of the links on that page are broken. NASA data archives and distribution are now managed by the Earth Observing System Data and Information System (EOSDIS) which uses a system of Distributed Active Archive Centers (DAACs). The Atmospheric Science Data Center (ASDC) is one of those DAACs. The format and delivery of ACEPOL data conform to the requirements of the ASDC, and are presented in a consistent manner with other field campaigns.*

*That said, we believe most the sentiment of user friendliness is a simple misunderstanding. Because of the impermanence of website URL's, we use Digital Object Identifiers (DOI's). If maintained properly, a DOI (using the website doi.org) will always point to the proper URL for a dataset. Use of DOI's is a requirement of this journal.*

*It appears the reviewer concatenated the URL and DOI's into non-existent URL's. For example, The ASDC archive can be accessed at:*
        *https://asdc.larc.nasa.gov/project/ACEPOL*
*OR by searching DOI.org for:*
        *10.5067/SUBORBITAL/ACEPOL2017/DATA001*
*But not at*
        *https://asdc.larc.nasa.gov/project/ACEPOL/10.5067/SUBORBITAL/ACEPOL2017/DATA 001*

*The DOI alone is most appropriate in the long term, since the URL may change.*

*This should resolve most of the reviewer's problems accessing data. We should also note that AERONET and CARB are externally collected and archived data that were not a part of the ACEPOL field campaign. ACEPOL flights targeted AERONET and CARB ground sites so that*

*data can be compared to those observations, but the structure of the archive is outside of our control. That said, data are relatively accessible given site name locations described in Table 4.*

**Comments:**
1. Calibrations: The calibrations effect, before and after the calibration of the observational data, is not clear. The details of flights are outlined in Table 4. Please provide results (in figures, etc) comparing raw data with the calibrated data.

*The calibration techniques vary considerably amongst the airborne instruments, are quite involved, and are not necessarily unique to the ACEPOL field campaign. For that reason, we cite the relevant literature for each instrument in the appropriate descriptive sections (3.1 and 3.2). Intercomparison of instrument observations referenced in Table 2 targets 1a-1c are the subject of ongoing research. When available, appropriate literature have been cited (e.g. Smit et al, 2019).*

2. What are the variables observed during the 9 flights? What are the time resolution of data of each observed variable?

*This is described in Table 1, and sections 3.1 and 3.2.*

3. Test of data accessibility from Table 3:

ASDC: https://eosweb.larc.nasa.gov/10.5067/SUBORBITAL/ACEPOL2017/DATA001
[screen capture page not found]

*(see above comments)*

AirMSPI
https://eosweb.larc.nasa.gov/project/airmspi/airmspi_table/10.5067/AIRCRAFT/AIRMSPI/ACEPOL/RADIANCE/ELLIPSOID_V006

[screen capture page not found]

*(see above comments)*

https://eosweb.larc.nasa.gov/project/airmspi/airmspi_table/10.5067/AIRCRAFT/AIRMSPI/ACEPOL/RADIANCE/TERRAIN_V006

[screen capture page not found]

*(see above comments)*

GroundMSPI :
https://eosweb.larc.nasa.gov/project/airmspi/airmspi_table/10.5067/GROUND/GROUNDMSPI/ACEPOL/RADIANCE_v009

[screen capture page not found]

*(see above comments)*

AERONET https://aeronet.gsfc.nasa.gov/

[screen capture for AERONET website]

Q: Where to find data relevant to the calibration of the 9 ER-2 flights described in Table 2?

*Relevant AERONET sites (observed for targets 2a and 2b) are described with their time in Table 4. Data are accessed on the site using that information. i.e. click the map, select the site, select date and time, etc.*

CARB https://www.arb.ca.gov/adam/index.html

[screen capture for CARB website]

Q: Where to get data relevant to 9 ER-2 flights in this work from this page?

*This is similar to AERONET: using the information in Table 4, one must search the CARB database*

4. Test of data accessibility from Abstract and reference ACEPOL Science Team, 2017:
doi:10.5067/SUBORBITAL/ACEPOL2017/DATA001
Test results: I was not able to find data from input above line to google.

*For DOI's it is best to use doi.org*

But I was managed to find data from following link:
https://asdc.larc.nasa.gov/project/ACEPOL/ACEPOL_AircraftRemoteSensing_CPL_ Data_1

[screen capture of ASDC website]

After trying to Get Dataset, with OPENDATA selected in Additional Options, following page poped up:

[screen capture of flight display and ordering system]

Even in here, it is still unclear where and how to view the flight data?

*This is a data ordering system, which gives the user the means to sort available data in time and space. You should have the option to download the data directly from that page, or order multiple granules of data. The (?) icon at the top right has more information on how this system works.*